# How does GPT-2 compute greater-than?: Interpreting mathematical abilities in a pre-trained language model

**Michael Hanna**[*]
ILLC
University of Amsterdam
m.w.hanna@uva.nl

**Ollie Liu**[*]
University of
Southern California
zliu2898@usc.edu

**Alexandre Variengien**[†]
Redwood Research
alexandre.variengien@gmail.com

## Abstract

Pre-trained language models can be surprisingly adept at tasks they were not explicitly trained on, but how they implement these capabilities is poorly understood. In this paper, we investigate the basic mathematical abilities often acquired by pre-trained language models. Concretely, we use mechanistic interpretability techniques to explain the (limited) mathematical abilities of GPT-2 small. As a case study, we examine its ability to take in sentences such as "The war lasted from the year 1732 to the year 17", and predict valid two-digit end years (years $> 32$). We first identify a circuit, a small subset of GPT-2 small's computational graph that computes this task's output. Then, we explain the role of each circuit component, showing that GPT-2 small's final multi-layer perceptrons boost the probability of end years greater than the start year. Finally, we find related tasks that activate our circuit. Our results suggest that GPT-2 small computes greater-than using a complex mechanism that activates across diverse contexts.

## 1 Introduction

As pre-trained language models (LMs) have grown in both size and effectiveness, their abilities have expanded to include a wide range of tasks, even without fine-tuning [6]. Such abilities can range from translation to text classification and multi-step reasoning [52]. Yet despite heavy study of these models [41, 42, 26], how LMs implement these abilities is still poorly understood.

In this paper, we study one such LM ability, performing mathematics. Mathematical ability has long been of interest in natural language processing: models have been trained to perform tasks such as simple arithmetic and word problems [51, 46]. Researchers have also fine-tuned pre-trained LMs on these tasks, instead of training from scratch [18, 25]. Recently, however, LMs seem to have acquired significant mathematical abilities without explicit training on such tasks [6, 34, 8, 16].

How these mathematical abilities arise in LMs is largely unknown. While studies have investigated pre-trained LMs' mathematical abilities [32, 38], existing work is behavioral: it explains *what* models can do, rather than *how* they do it. Most work that delves into model internals does so using models trained directly on such tasks: Hupkes et al. [23] probe such models for hierarchical structure, while Liu et al. [27] and Nanda et al. [31] study toy models trained on modular addition. Some studies do examine the structure of number representations in pre-trained models [30, 49]; however, they do not provide a causal explanation about how these models leverage these representations to perform math. The mechanisms underlying pre-trained LMs' mathematical abilities thus remain unclear.

---

[*]Work performed as part of Redwood Research's REMIX program
[†]Work performed during an internship. Now at Conjecture

37th Conference on Neural Information Processing Systems (NeurIPS 2023).

Figure 1: Year-span prediction example (`XX`=17 and `YY`=32) with sample (in)valid output years.

To understand the roots of these mathematical abilities, we study them in GPT-2 small,[1] [39] which we show still possesses such abilities, despite its small size. This small size enables us to investigate its mathematical abilities at a very low level. Concretely, we adopt a circuits perspective [35, 14], searching for a minimal subset of nodes in GPT-2's computational graph responsible for this ability. To do so, we use fine-grained, causal methods from mechanistic interpretability, that allow us to identify nodes in GPT-2 that belong in our circuit, and then prove our circuit's correctness, through carefully designed causal ablations [21]. We also use mechanistic methods to pinpoint how each circuit component contributes to the mathematical task at hand. The end result of this case study is a detailed description of GPT-2's ability to perform one simple mathematical operation: greater-than.

Our investigation is structured as follows. We first define year-span prediction, a task that elicits mathematical behavior in GPT-2 (Section 2). We give the model input like "The war lasted from the year 1732 to the year 17"; it assigns higher probability to the set of years greater than 32. We next search for the circuit responsible for computing this task, and explain each circuit component's role (Section 3). We find a set of multi-layer perceptrons (MLPs) that computes greater-than, our operation of interest. We then investigate how these MLPs compute greater-than (Section 4). Finally, we find other tasks requiring greater-than to which this circuit generalizes (Section 5).

Via these experiments, we accomplish two main goals. First, we find a circuit for greater-than in GPT-2. This brings new mechanistic insights into math in pre-trained LMs, and builds on the limited existing work on circuits in pre-trained LMs [50] by examining a new task with a wide output space and rich structure. Second, we show that GPT-2's greater-than relies on a complex circuit that activates across contexts. This mechanism surpasses simple memorization, but does not reflect full mathematical competence; it lies between memorization and generalization. We thus add nuance to the memorization-generalization dichotomy, and take the first step towards a rich characterization of the states in between them.

## 2   Year-Span Prediction in GPT-2

GPT-2's size is ideal for low-level study, especially with potentially resource-intensive techniques like those in Section 3.1. However, this small size poses a challenge: GPT-2 is less capable than larger LMs, which still often struggle with mathematical tasks [29]. With this in mind, we craft a simple task to elicit a mathematical behavior in GPT-2, and verify that GPT-2 produces said behavior.

**Task and Dataset**   We focus on a simple mathematical operation, greater-than, framed as it might naturally appear in text: an incomplete sentence following the template "The `<noun>` lasted from the year `XXYY` to the year `XX`" (Figure 1). The model should assign higher probability to years >`YY`. We automatically generate sentences using this template. We draw the nouns from a pool of 120 nouns that could have a duration, found using FrameNet [1]; Appendix G lists the full pool of nouns. We sample the century `XX` of the sentence from $\{11, \ldots, 17\}$, and the start year `YY` from $\{02, \ldots, 98\}$.

We impose the latter constraints because we want GPT-2 to be able to predict a target as it would naturally be tokenized. However, GPT-2 uses byte-pair encoding, in which frequent strings more often appear as single tokens [44]. Thus, more frequent years—multiples of 100 or those in the 20th century—are tokenized as single tokens; less frequent years are broken into two. This causes a problem: GPT-2 could predict "[00]" after "[17]', but "1700" is always tokenized as "[1700]" in normal data and never as "[17][00]". So, we exclude all single-token years from our year pool. Finally, we want each example to have at least one correct and one incorrect validly tokenized answer, so we exclude each century's highest and lowest validly tokenized year from the pool of start years.

---

[1]Further references to GPT-2 refer to GPT-2 small

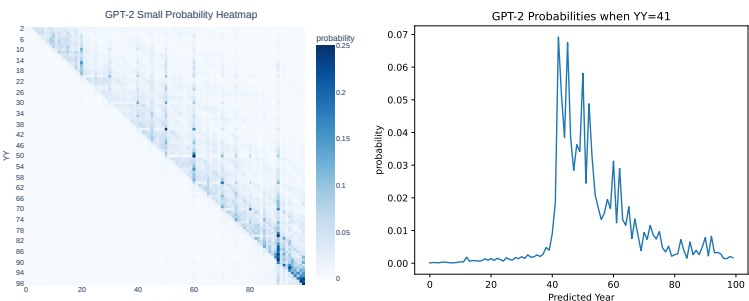

Figure 2: Left: Probability heatmap of GPT-2 for year-span prediction. Y-axis: the sentence's start year (YY). X-axis: the two-digit output year candidate. (X,Y): Mean probability assigned by GPT-2 to output year X given input year Y. Right: GPT-2's average output distribution when YY=41

**Qualitative Evaluation**    We first qualitatively analyze GPT-2's baseline behavior on this task by running it on a dataset of 10,000 examples. Each example has a noun randomly drawn from our 120 nouns, and a year drawn randomly from the 768 valid years from 1000 to 1899. For each YY in $\{2, \ldots, 98\}$, we take the average of GPT-2's probability distribution over predicted years, for all examples with start year YY; we visualize these average distributions in Figure 2.

GPT-2 appears to perform greater-than on our year-span prediction task: it creates a sharp cutoff between invalid end years ($\leq$ YY), and valid end years ($>$ YY). It assigns higher probability to the latter years, though not all of them: 15-20 years after YY, probabilities drop. The exact length of the year-span receiving higher probability likely reflects patterns in GPT-2's training data. In the real-world, and likely also in GPT-2's training data, our prompts' nouns, such as a "war," "dynasty", or "pilgrimage", have average durations that GPT-2 may have learned, influencing its output.

**Quantitative Evaluation**    We design two numerical measures of model performance for the purpose of quantitative assessment. Let YY $\in \{02, \ldots, 98\}$ be the start year of our sentence, and $p_y$ be the probability of a two-digit output year $y$. We define the following two metrics:

- **Probability difference**: $\sum_{y > \text{YY}} p_y - \sum_{y \leq \text{YY}} p_y$
- **Cutoff sharpness**: $p_{\text{YY}+1} - p_{\text{YY}-1}$

Probability difference verifies that model output reflects a greater-than operation by measuring the extent to which GPT-2 assigns higher probability to years $>$YY. It ranges from -1 to 1; higher is better. In contrast, cutoff sharpness is not intrinsically connected to greater-than. However, it quantifies an interesting behavior of GPT-2: the sharp cutoff between valid and invalid years. In doing so, it checks that the model depends on YY, and does not produce constant (but valid) output, e.g. by always outputting $p(99) = 1$. Cutoff sharpness ranges from -1 to 1; larger values indicate a sharper cutoff.

We perform this evaluation with our same 10,000-element dataset; on this dataset, GPT-2 achieves 81.7% probability difference (SD: 19.3%) and a cutoff sharpness of 6.0% (SD: 7.2%). Overall, both qualitative and quantitative results indicate that GPT-2 performs the greater-than operation on the year-span prediction task. For more study of GPT-2's behavior on this task, see Appendix A.

## 3    A Circuit for Year-Span Prediction

Having defined our task, we now aim to understand how our model performs it internally. Since the advent of pre-trained models, many methods, such as attention analysis [9] and probing [3], have sought to answer this question. Probing, which trains auxiliary models (probes) to extract information from model representations, has been particularly popular; it has been used to localize syntactic and semantic processing in many pre-trained LMs [37, 45, 15]. However, it has significant pitfalls: most crucially, probes can sometimes extract information from model representations that is irrelevant to model behavior [13, 40, 22]. To avoid this issue, other work has used causal interventions [36] to intervene on model internals, and observe changes in model behavior [20, 48, 17]. This ensures that our insights about model internals are actually functionally relevant to model behavior.

In light of this, we examine GPT-2's task performance by using causal techniques to identify a **circuit**: a minimal computational subgraph of our model that suffices to compute the task [35, 14]. While many causal methods aim to identify important components (nodes) of models [48, 11, 17], the circuits methodology instead considers important *edges*. Circuits thus examine not only important nodes, but also their interactions, and how they work together to support model behavior.

Below, we explain the *path patching* technique, and how to use it to find circuits (Section 3.1). We then find a circuit for greater-than, and prove its correctness (Section 3.2). Finally, we assign semantics to the nodes and edges of our circuit (Section 3.3). All of our experiments use the `rust-circuit` library, and our code is available at `https://github.com/hannamw/gpt2-greater-than`. For information on how to apply this methodology to other problems, see Appendix H.

## 3.1 Path Patching

To find a circuit, we use path patching, introduced by Wang et al. [50] and further described by Goldowsky-Dill et al. [21]. This technique determines how important a model component (e.g. an attention head or MLP) is to a task, by altering that component's inputs and observing model behavior post-alteration. It is much like causal mediation analysis or interchange interventions [48, 17]; however, unlike these, it allows us to constrain our intervention's effects to a specific path.

To illustrate this, consider a model's forward pass on its inputs as a directed acyclical graph. Its nodes are components such as attention heads or MLPs. The input of a node $v$ is the sum of the outputs of all nodes with a direct edge to $v$. GPT-2 can be thought of as such a graph flowing from its input tokens to its logits (and thereafter, its probabilities), as depicted in Figure 3.

In path patching, we specify new input tokens, and a path of components through which they will reach the logits. For example, if we want to ascertain the effects of MLP 10 on the logits, we can patch the direct path (MLP 10, logits) with new input, which we call the 01-input: "The war lasted from the year 1701 to the year 17". We thus alter MLP 10's direct effects on the logits without changing its output to the attention and MLP of layer 11 (Figure 3). If the model's behavior (as indicated by its logits) changes, we can be sure that this is because MLP 10 is important to that behavior; it is not due to downstream components. Earlier methods like interchange interventions lack this distinction—when they alter a component, they affect all components downstream from it.

The specificity of path patching allows us to test detailed hypotheses. For example, imagine that we know that MLP 10 affects the logits both directly and via its effects on MLP 11. We want to know how important layer 10's attention is to the circuit via MLP 10. We can test this by patching two paths at once: (Attn 10, MLP 10, logits) and (Attn 10, MLP 10, MLP 11, logits), as in Figure 3. This allows us to pinpoint the relationship between precisely these two components, Attn 10 and MLP 10. This technique underpins our circuits approach: we search for a path starting in the inputs and ending in the logits that explains how our model performs the greater-than task.

To perform path patching, we need a new dataset that replaces a node's original inputs. To this end, we create the "01-dataset": we take each example in the original dataset and replace the last two digits YY of the start year with "01". If a component normally boosts logits of years>YY, patching it with the 01-dataset will cause it to boost the logits of years $> 01$, inducing a larger error in the model.

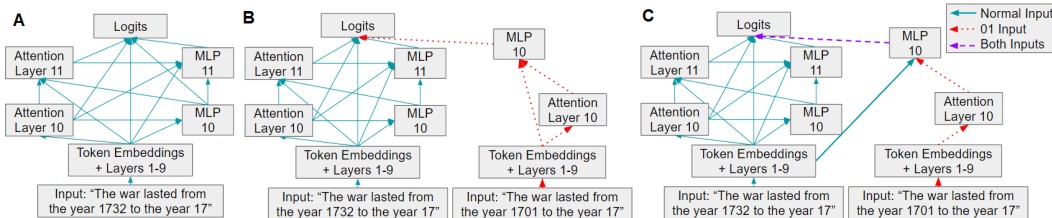

Figure 3: A. The computational graph of GPT-2, run on our normal dataset. B: GPT-2, where the (MLP 10, logits) path is patched to receive 01-input. C. GPT-2, where the (Attn 10, MLP 10, logits) path receives 01-input. Nodes receiving normal input have blue output; nodes receiving 01-input have red output; nodes receiving both have purple output. Note that in B and C, there is no longer an edge connecting MLP 10 and the logits.

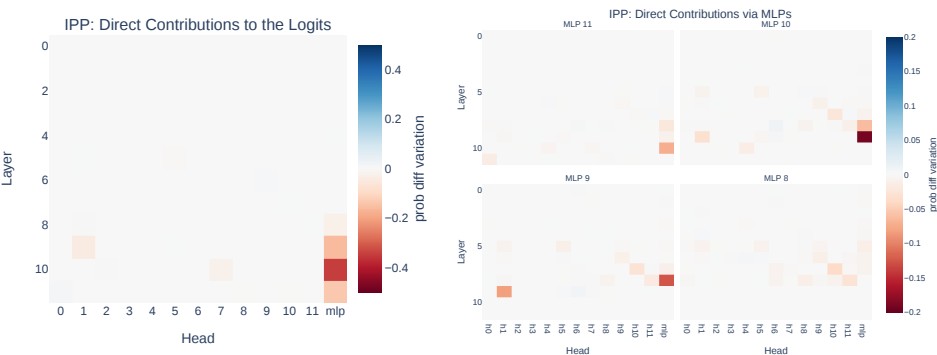

Figure 4: Iterative path-patching (IPP) heatmaps. Y-axis: layer of the component. X-axis: attention head number, or MLP. (X,Y): Change in probability difference induced by patching the corresponding component. A: Heatmap for the path ((X,Y), logits). B: Heatmaps for MLPs 8-11.

## 3.2 Circuit Components

**MLPs**  We search for a circuit by identifying components that perform year-span prediction via their direct connection to the logits. We consider as potential components GPT-2's 144 attention heads (12 heads/layer×12 layers), and 12 MLPs (1 per layer). We do so because the residual stream [14] that serves as input to the logits is simply the sum of these components' direct contributions (along with the token embeddings; we ignore these as they contain no YY information). If we consider each of these, we will not miss any components that contribute to this task. For details, see Appendix C.

We iteratively path patch each component's direct contributions to the logits, replacing its inputs with the 01-dataset. In our earlier notation, for a component of interest $C$, we patch the path ($C$, logits), as in Figure 3 B, where $C$ = MLP 10. We patch only one component at a time, and only at the end of the sentence; at other positions, these components cannot affect the logits directly.

After we patch a component, we run the model and record the probability difference, comparing it to that of the unpatched model. If patching a component caused model performance to change significantly, that component contributed to the model's computation of year-span prediction.

Figure 4 shows the results of this experiment; for computational reasons we run it using a smaller dataset (490 datapoints, 5 per year YY). The heatmap indicates that MLPs 8-11 are the most important direct contributors to the logits, along with a9.h1: attention layer 9's head 1. However, the MLPs cannot act alone: to compute years>YY, these MLPs at the end of the sentence must know the value of YY. But unlike attention heads, MLPs cannot attend to earlier tokens such as the YY token. Thus, we search for nodes that contribute to the circuit via these MLPs.

**Attention Heads**  We find components that contribute to the circuit via the MLPs using more path patching. We start by patching components through MLP 11, since it is the furthest downstream; for a component of interest $C$, we patch ($C$, MLP 11, logits). We find that MLP 11 relies mostly on the 3 MLPs upstream of it (Figure 4), so we search for components that act via those MLPs.

We next find components that contribute to the circuit through MLP 10. For a given $C$, we patch ($C$, MLP 10, logits) and ($C$, MLP 10, MLP 11, logits), as in Figure 3 C. We do so because MLP 10 contributes directly to two nodes in our circuit, the logits and MLP 11, and we want to know which nodes contribute via MLP 10 to the entire circuit. We repeat this procedure for MLPs 9 and 8.

The results in Figure 4 indicate that MLPs rely heavily on other MLPs upstream of them. MLPs 8 and 9, the furthest upstream of our MLPs, also rely on attention heads. MLP 9 relies on a9.h1, while MLP 8 relies on a8.h11, a8.h8, a7.h10, a6.h9, a5.h5, and a5.h1; we add these to our circuit. Many of these attention heads can be seen to contribute to the logits directly, though more weakly than the MLPs do. For this reason, we also add these heads' direct connections to the logits to our circuit.

Figure 5 visualizes the circuit we have found. We could further develop this by specifying a circuit from the token inputs to the logits; indeed, we do so in Appendix B. However, the present circuit

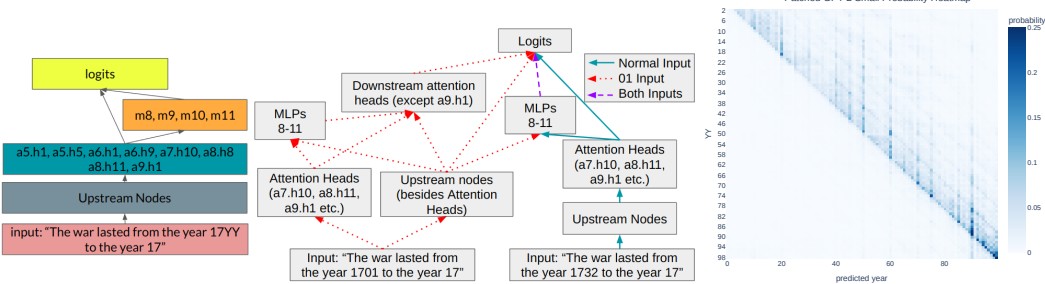

Figure 5: Left: Diagram of the year-span prediction circuit. Center: Diagram showing which GPT-2 components receive our standard dataset vs. our 01-dataset in the circuit evaluation experiment. Right: The probability heatmap (as in Figure 2) for the patched model.

already captures the most interesting portion of the model: the MLPs that compute greater-than. So, we instead provide evidence that our circuit is correct, and then analyze its constituent parts.

**Evaluation**   Having defined our circuit, we perform another path-patching experiment to ensure it is correct. In this experiment, we give most of the model inputs from the 01-dataset. The model only receives our standard dataset via the paths specified in our circuit. So, our attention heads' contributions to the logits are backed by the standard dataset, as are their contributions to the MLPs, and the MLPs' contributions to one another. But, some components of the MLPs' inputs (those that come from model components not in the circuit) receive input from the 01-dataset as well. We stress that this is a difficult task, where the large majority of the model receives input that should push it to poor performance. For a diagram of the circuit and our evaluation, see Figure 5.

We perform this evaluation using the larger dataset, and almost entirely recover model performance. The probability difference is 72.7% (89.5% of the original) and the cutoff sharpness is 8%—sharper than pre-patching. This indicates that our circuit is mostly sufficient to compute this task. The circuit is also necessary: performing the opposite of the prior experiment, giving nodes in our circuit the 01-dataset, and those outside it the normal dataset, leaves GPT-2 unable to perform the task: it achieves a probability difference of -36.6%.

If we target other circuits of a size and location similar to our circuit's, performance is related to the preservation of the paths from the input, to our attention heads, our MLPs (especially MLPs 9 and 10), to the logits. If the paths are interrupted (i.e. no attention head or no MLP overlap with the original circuit), performance is very low. If at least one path is preserved, performance improves with each additional component in common with the original circuit; MLPs have the biggest impact.

## 3.3   Circuit Semantics

Now, we interpret each circuit component, starting with the attention heads. We first perform a simple attention-pattern analysis of the heads in our circuit. Figure 6 shows which tokens our attention heads attend to at which positions. At the relevant (end) position, in-circuit attention heads attend to the YY position, suggesting that they detect the year which the output year must be greater than.

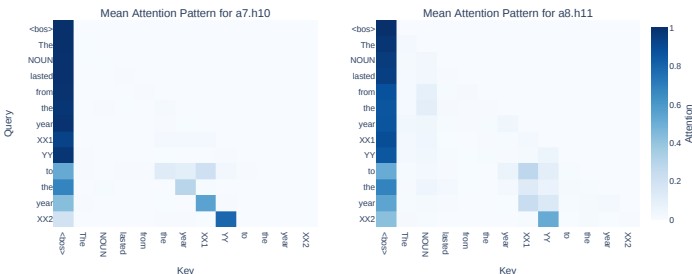

Figure 6: Attention patterns of a7.h11 and a8.h10. <bos> denotes GPT-2's start of sentence token.

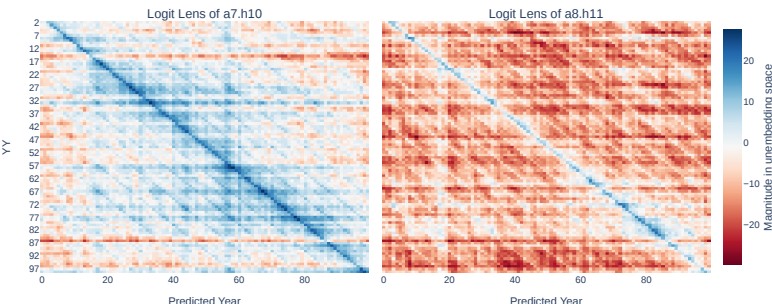

Figure 7: Logit lens of a7.h11 and a8.h10. Axes as in Figure 2; blue indicates that the module upweights the output year, and red, that it downweights the year.

Next, we examine the contributions of attention heads using the logit lens approach [33]: we multiply each head's output by GPT-2's unembedding matrix, translating this output into unembedding (vocabulary) space. Note that here, we do not only view logit lens as a tool for obtaining intermediate estimates of model predictions [4]. Rather, we also use it to understand components' outputs more generally: the logit lens can capture how such outputs shape model predictions, but it can also capture how these outputs add information to the residual stream in unembedding space.

We visualize the heads' outputs for each sentence in our small dataset in Figure 7. Attention head outputs for a sentence with start year YY have a high dot product with the embedding vector for YY, as shown by the blue diagonal in the plots; this indicates that the head upweights YY, making it a more likely output. Note that YY was not just upweighted highly compared other 2-digit years; YY was the most highly upweighted token across all tokens. Given our earlier analysis, we therefore hypothesize that these attention heads identify the start year (at the YY position), and indicate it via a spike in unembedding space of the residual stream at the end position; they thus communicate YY to downstream components.

We similarly apply the logit lens to the outputs of MLPs 8-11 (Figure 8). The results indicate that MLPs of 9 and 10 directly specify which years are greater than YY: the logit lens of each layer's output has an upper triangular pattern, indicating that they upweight precisely those years greater than YY. MLP 11 plays a similar role, but seems to upweight roughly the first 50 years after YY, enforcing a maximum duration for the event in the sentence. However, MLP 8 is unusual: its logit lens shows a diagonal pattern, but no upper triangular pattern that would indicate that it computes greater-than.

We claim that this is because MLP 8 contributes mainly indirectly, via the other MLPs in our circuit. We confirm this by patching MLP 8's direct contributions to the logits with the 01-dataset; we do so again with its indirect contributions, through the other MLPs. In the former case, model performance drops only 14%, while in the latter case, it drops by 39%. So MLP 8 does not contribute much to the logits directly, but it does contribute indirectly. Other MLPs also have mixed effects: MLP 9 has roughly equal direct and indirect contributions (28% vs. 32%), while MLP 10 contributes mostly directly (56% vs. 16%). MLP 11 can only contribute directly.

Our full picture of the circuit so far is this: the attention heads communicate the start year YY in embedding space. MLP 8's mostly influences downstream MLPs. However, MLPs 9, 10, and 11 appear to compute the greater-than operation in tandem, and in steps. We conclude that while the attention heads identify the important year YY, it is the MLPs that effect the greater-than computation.

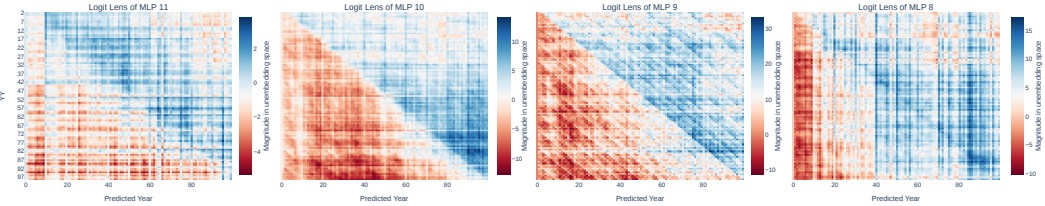

Figure 8: (Left to right) Logit lens of MLPs 11, 10, 9, and 8; labels as in Figure 7

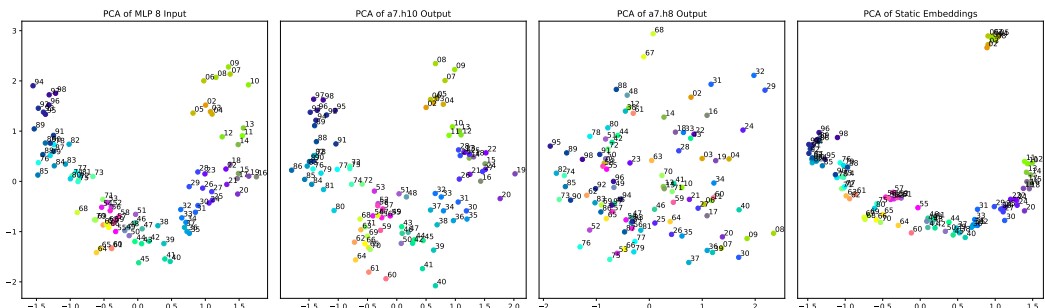

Figure 9: PCA of MLP 8's input, a7.h10's and a7.h8's output, and the static year embeddings. Each point corresponds to one datapoint's representation, and is labeled with and colored by the its YY.

# 4 Explaining Greater-Than in the Year-Span Prediction Circuit

Our prior experiments show that MLPs 9-11 directly compute greater-than. But how do they do so? We cannot provide a conclusive answer, but identify avenues by which MLPs might compute this. We first examine their inputs, finding structure that might enable greater-than computation. Then, we examine MLP internals, showing how neuron composition could enable greater-than computation.

## 4.1 Input Structure

To understand how MLPs compute greater-than, we analyze various model representations using 2D Principal Component Analysis (PCA). For each of the 97 datapoints in our small dataset, each with a unique start year, we analyze the input residual stream to our MLPs, as well as the output of relevant attention heads. As a control, we also analyze representations from irrelevant model components, and the static year embeddings. We take all component representations from the end position.

In Figure 9, PCA reveals that the input residual stream to MLP 8 (and indeed all of our MLPs, though not all are shown) is highly structured: representations are ordered by the start year of the sentence they are from, increasing clockwise. The same is true of the outputs of relevant attention heads (a7.h10), which serve as inputs to the MLPs, but not of outputs of irrelevant heads (a7.h8). This suggests that it is specifically the relevant attention heads that transmit this structured information to relevant MLPs. But while the heads seem to transmit this structured information to the MLPs, they need not have created this structure from scratch. We find, as in Wallace et al. [49], that structure already exists in the static year embeddings, though the years 02-09 are clustered apart from the rest. The heads need only unify these groups and move this information from the YY position to the end.

Structured number representations have been implicated in mathematical capability before: Liu et al. [27] train a toy transformer model on modular addition, and find that its number representations become structured only after it stops overfitting and begins to generalize. This suggests that GPT-2's structured number representations may be relevant to its greater-than ability. However, our experiments struggle to prove this causally. When we ablate the dimensions found through PCA, to test their importance to the greater-than task, we found little change in model performance. Similarly, removing linearly-extractable YY information from attention head output using LEACE [5] yielded only slightly lower probability difference (64.7%) than the baseline (81.7%). This indicates that structured YY information may not fully explain GPT-2's greater-than abilities.

## 4.2 Neuron-Level Processing

To better understand MLPs, we turn to their internals, zooming in on their neurons. We choose to study MLP 10 closely, as we know it directly contributes to the greater-than operation. We start by asking which of MLP 10's neurons are important—a sort of question already studied using probing [10, 12], causal ablations [2, 24, 48], and other techniques; see Sajjad et al. [43] for an overview.

As before, we use path patching because it provides precise causal insights. We path patch each of MLP 10's 3072 neurons' direct contributions to the logits with the 01-dataset. We again record the change in model performance, as measured by probability difference, compared to the unpatched

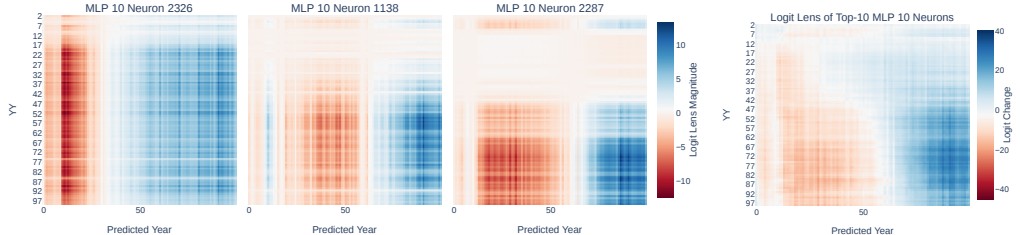

Figure 10: Left: The logit lens of the 3 MLP 10 neurons most important to year-span prediction. Right: The logit lens of the top-10 MLP 10 neurons. Blue indicates that the neuron upweights logits at the given input year (y-axis), output year (x-axis) combination, while red indicates downweighting.

model. We find that neuron contributions to the task are sparse: most neurons can be patched (ablated) with near zero effect on our model performance, as observed in prior work [48].

We then analyze those neurons that contribute most to model performance using the logit lens. To do this, we take advantage of the fact that each neuron has a corresponding row in the MLP output weight matrix. As noted by Geva et al. [19], multiplying this row by the unembedding weights yields an (unnormalized) distribution over the logits, indicating which outputs the neuron upweights when activated. Taking the outer product of this logit distribution with the neuron's activations yields the logit lens, indicating which output years the neuron upweights for each input sentence's YY.

Figure 10 shows the logit lens of the 3 most important neurons in MLP 10; more neurons can be found in Appendix D. Each neuron up- or down-weights certain output years depending on the input year YY, but no individual neuron computes greater-than. No one neuron can do so, as each neuron's activation for each input only scales that neuron's distribution over the logits, without changing its overall shape. In contrast, the correct shape of the logits differs depending on the example's start year.

Many neurons can compute greater-than when combined, though. We perform logit lens on the sum of the top-10 neurons' contributions [2] in Figure 10. Though they do not do so individually, the top-10 neurons perform an imperfect greater-than when summed together as a group. The logit lens of MLP 10 as a whole can be thought of as the logit lens of the sum all 3072 neurons' contributions; we partially recreate this with the contributions of just the top-10 neurons. Including more top neurons produces sharper approximations of greater-than; see Appendix D for examples.

In summary, we find that even inside one MLP, the greater-than operation is spread across multiple neurons, whose outputs compose in a sum to form the correct answer. Even the contributions of a small number of relevant neurons composed begin to roughly form the correct operation. We study this in MLP 10, but observe it in the other MLPs as well. Section 3.3 suggested that GPT-2 computed greater-than across multiple important MLPs; these results suggest that moreover multiple important neurons in each MLP compose to allow the MLP to compute greater-than.

## 5  Does The Circuit Generalize?

We now possess a detailed circuit for year-span prediction. But one thing remains unclear: is this a general circuit for the greater-than operation? Or does it only only apply to this specific, toy task? Answering this question fully would require an in-depth exploration of circuit similarity that fall outside the scope of this paper. For the purposes of this investigation, we perform primarily qualitative analyses of tasks that preserve the format and output space of year-span prediction.

To start, we focus on three increasingly different prompts: "The <noun> started in the year 17YY and ended in the year 17", "The price of that <luxury good> ranges from $ 17YY to $ 17", and "1599, 1607, 1633, 1679, 17YY, 17". In all cases, a two-digit number greater than YY would be a reasonable next token. The model performs greater-than given all of these prompts ($\geq 69\%$ probability difference). Moreover, the circuits found via path patching are similar to those in Sections 3.2 and 3.3. When we run GPT-2 on the first two tasks, ablating all edges not in our original year-span circuit, we achieve 98.8% and 88.9% loss recovery respectively; for more details, see Appendix F. As before,

---

[2]We can confirm this using path patching as well; see Appendix E for details.

these tasks depend on MLPs 8-11 to compute greater-than; these MLPs depend on attention heads that transmit information about YY.

That said, these tasks' circuits are not all identical to the greater-than circuit. GPT-2 recovers only 67.8% of its original performance on the last when ablating everything but the year-span circuit. But, a closer look at path patching results indicated that on this task, GPT-2 used not only the entire year-span circuit, but also MLP 7 and two extra attention heads. After including those nodes, GPT-2 recovered 90.3% of performance. Similar tasks seem to use similar, but not identical, circuits.

GPT-2 produced unusual output for some tasks requiring other mathematical operations. It produced roughly symmetric distributions around YY on the task "1799, 1753, 1733, 1701, 16YY, 16", which might yield predictions smaller than YY. It behaved similarly on examples suggesting an exact answer, such as "1695, 1697, 1699, 1701, 1703, 17", which could yield 05. GPT-2 even failed at some tasks that were solvable using the greater-than circuit, like "17YY is smaller than 17"; it always predicted YY. Across all such tasks, we found that GPT-2 relied on another set of heads and MLPs entirely. So GPT-2 does not use our circuit for all math; sometimes it does not rely on it even when it should.

We also observe the opposite phenomenon: inappropriate activation of the greater-than circuit, triggered by prompts like "The <noun> ended in the year 17YY and started in the year 17" and "The <noun> lasted from the year 7YY BC to the year 7". In these cases, GPT-2 ought to predict numbers smaller than YY; however, it predicts numbers greater than YY. This is because it is using the exact same circuit used in the greater than case! GPT-2 thus overgeneralizes the use of our circuit.

Our results suggest that our circuit generalizes to some new scenarios. But what does a generalizing circuit imply about the origins of GPT-2's greater-than capabilities—do they stem from memorization [47, 7], or rich, generalizable representations of numbers [27]? Our complex circuit and the structured numbered representations found in our PCA experiments hint at some mathematical knowledge in GPT-2. However, the presence of a greater-than circuit does not preclude memorization. Our circuit could function internally as a lookup table, where attention heads transmit YY information to MLPs that then upweight the years that they have memorized to be >YY. In this case, (incorrect) generalization would involve GPT-2 (incorrectly) activating the lookup circuit based on context.

Though our current evidence does not allow us to definitively attribute GPT-2's behavior to generalized mathematical ability or memorization, it suggests that the underlying mechanism is something in between the two. The lack of causal evidence for the role of structured number representations, and the fact that GPT-2 cannot handle related operations like "less-than" or "equal-to" argue against generalized math mechanisms. However, even if we view this circuit as simply retrieving memorized facts, the retrieval mechanism at work is sophisticated. GPT-2 can (imperfectly) identify greater-than scenarios and the relevant operand; it then activates a dedicated mechanism that retrieves the correct answer. GPT-2's mathematical abilities thus extend beyond a simple, exact memorization of answers.

## 6 Conclusion

In this paper, we attempted to bridge the gap between our understanding of mathematical abilities in toy models, and the mystery of such abilities in larger pre-trained LMs. To do so, we outlined a circuit in GPT-2 with interpretable structure and semantics, adding to the evidence that circuits are a useful way of understanding pre-trained LMs, [50] even in a more complex scenario. Our circuit is coarser-grained than findings in toy models for mathematical tasks [31], but much finer-grained than existing work on mathematics in pre-trained LMs. Moreover, we showed that this circuit in GPT-2 activates across contexts on other greater-than-adjacent tasks. Whether such cross-context activation reflects generalization or memorization is an open question for circuits work in general.

We note that our conclusions are limited by the small size of our model and dataset, and the simple phenomenon studied. Our study is very model-centric: data-driven interpretability techniques would strengthen our work. Studying circuit performance across diverse tasks could better measure the degree to which our circuit generalizes to all greater-than tasks. Similarly, studying larger models would confirm that our results hold for the models that dominate natural language processing today.

Despite these limitations, we believe that this study lays the groundwork for future work. Our small study hints at the potential for circuits as a lens for the study of memorization and generalization in pre-trained LMs. More broadly, we hope that our finding that not only attention heads, but also MLPs and their neurons can be analyzed jointly as a complex system will motivate circuits work to come.

## Acknowledgments and Disclosure of Funding

The authors thank Buck Shlegeris, Chris MacLeod, and Arthur Conmy for their valuable feedback on earlier drafts of this work. They also thank Neel Nanda for a useful research meeting about this work. They additionally thank Dani Yogatama, as well as members of the Amsterdam and Technion NLP groups, for their helpful comments. They also appreciate the insightful reviews and discussion provided by the anonymous reviewers for this paper. They finally thank Redwood Research for both running the REMIX program, which provided many of the ideas, techniques, and collaborations behind this paper, and providing continued computational and travel support thereafter.

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

# A  More Behavioral Study of Greater-Than in GPT-2

In this section, we discuss the results of additional behavioral studies of GPT-2's behavior.

**GPT-2 also predicts valid XX**    Although our previous experiments show that GPT-2 predicts a valid two-digit end year YY, we also verify that GPT-2 predicts a valid continuation when the output prefix XX is not provided. In this scenario, GPT-2 should either predict either a two-digit century token whose value is $\geq$ XX, or an entire 4-digit year token that is $\geq$ XXYY. We test this using the same dataset as in previous experiments, with the final XX removed. We find that on average, considering the top-100 tokens (95% of probability mass), 94% of their probability mass (89% of the total) is assigned to valid continuations. So, GPT-2 generally predicts valid XX.

**GPT-2's top predictions are valid**    As suggested by the high probability difference achieved by GPT-2, its top YY predictions for an input from our dataset are quite good. 100% of the top-1 continuations, and 98.6% of top-5 continuations were correct.

**Default Behavior in GPT-2**    GPT-2 predicts the greater-than in greater-than scenarios, but also in scenarios where less-than would be appropriate (see Section 5). For this reason, we also tested how GPT-2 behaves in a scenario where neither greater-than nor less-than in required, in order to a elicit a sort of default behavior.

To test this, we created sequences of 4-digit numbers like "XXY1, XXY2,..., XXYN, XX". We chose XX, Y1,...,YN randomly for each sequence, respecting tokenization. Sequences were generally not monotone, so any 2-digit continuation YY could have been valid. We found that GPT-2's behavior depends on the sequence length $N$. At low $N$, GPT-2 produces mostly numbers > YN; the proportion of continuations < YN increases smoothly until 50% by $N = 20$. So even in the context of random sequences, GPT-2 often generates increasing numbers.

Beyond simple behavioral study, we conducted a circuits analysis on this task, and found that our greater-than circuit also underlies this behavior. This provides another example of GPT-2's ability to identify greater-than situations is flawed. However, it also demonstrates again that whenever we see this greater-than behavior, our circuit is responsible. An open question worth answering is why our circuit activates in such incorrect scenarios, and if this can be mitigated.

# B  The Full Year-Span Prediction Circuit

Now, we describe the rest of the year-span prediction circuit. This is actually not very large, as we already know most of the important components. All that remains is to understand how the input to the attention heads is crafted.

We can investigate this via iterative path-patching again: we will look for nodes that influence the attention heads. This can be done in three ways: via queries, keys, and values. The queries and keys jointly determine what the attention heads attend to. In theory, attention patterns should be relatively constant across examples: in all cases, attention heads should attend to the YY position. If this is the case, we should be able to ignore the queries and keys, and focus only the value.

In practice, attention patterns are not exactly constant as YY changes. While broad trends are similar, the intensity of the attention to the YY position varies (though not linearly with YY, as might make sense: see Figure 11). At YY=01, attention to the YY position is rather low, meaning that model greater-than behavior is less pronounced when patched. Thus, the queries and keys are somewhat important: patching them with bad data reduces performance by 15% with respect to our partial circuit. The influences on these heads via the keys are similar to those on the values, discussed in the next paragraph. The influences via the queries are distinct, but we will set these aside, and focus on the value vectors.

The most important influences on these heads are the influences on their values at the YY position. The values are combined to form each attention head's output: patching the values with 01-input entirely disrupts circuit performance, unlike patching its keys or queries.

To find influences on these values, we iteratively path patch potential components that might communicate with our attention heads via their values at the YY position. We find (Figure 12) that these

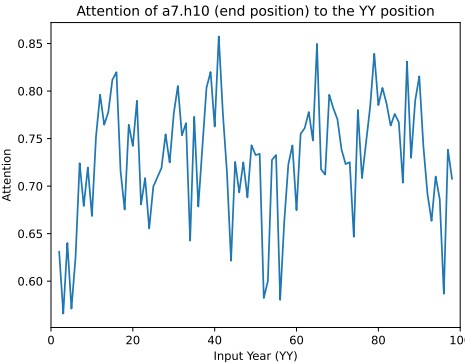

Figure 11: Attention from a7.h10 (end position) to the YY position, by input year YY

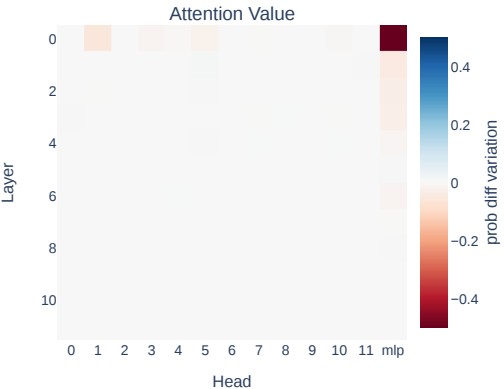

Figure 12: Iterative path patching results through attention heads' value vectors

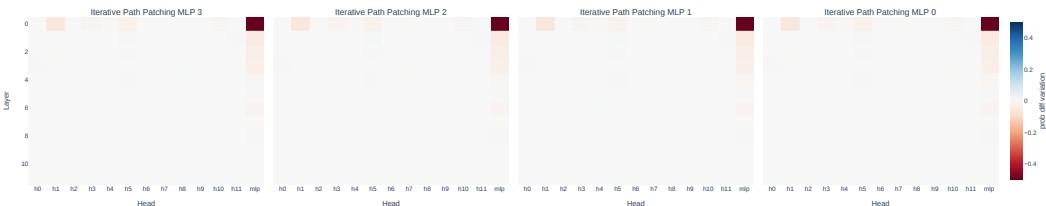

Figure 13: Iterative path patching results for MLPs 0-3

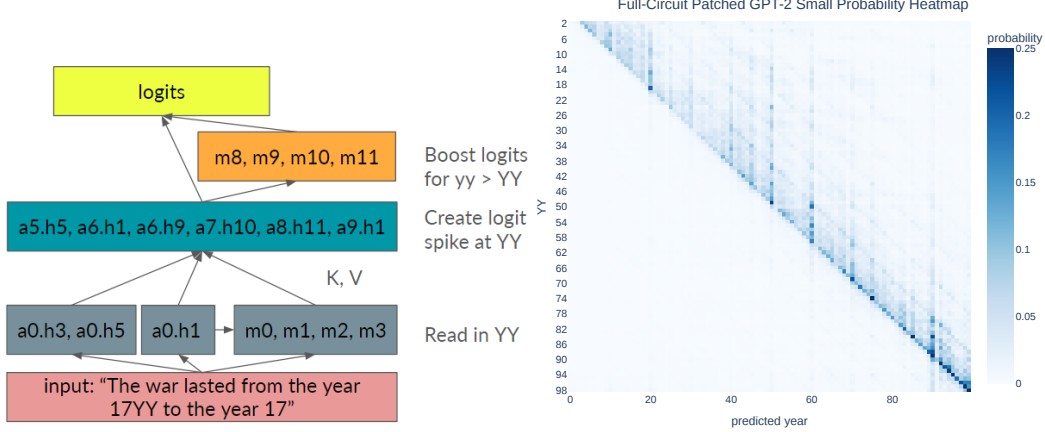

Figure 14: (Left) Full circuit diagram. Note that grouped MLPs are interconnected; attention heads are not. (Right) Probability heatmap for the patched full circuit.

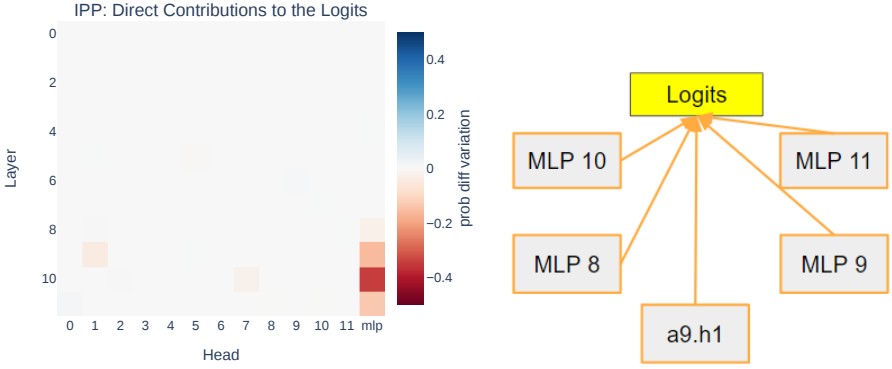

Figure 15: Path Patching Step 1: Logits

are mostly MLPs 0-3, as well as a0.h5, a0.h3, and a0.h1. We delve again into this group of MLPs (Figure 13), and see that each MLP relies on the MLPs before it and a0.h1; MLP 2 is the exception, as it does not rely on MLP 1. We know that a0.h5 can only rely on the token embeddings; there is nothing else before it in the residual stream! We attempt to find what MLP 0 depends on, but it does not rely on any of the attention heads prior to it; this indicates that it depends primarily on the token embeddings, not shown in these iterative path patching diagrams.

We have now developed a hypothesis regarding a full circuit, pictured in Figure 14. We evaluate it as before, keeping in mind that the keys and values of the attention heads are patched with the same input components as found earlier; the queries, not the focus of this section, receive all good inputs. The circuit achieves a probability difference of 71.5% (98.3% of what we achieved earlier), and a cutoff sharpness of 10.5% (again sharper than pre-patching). The qualitative results are in Figure 14.

## C    Circuit Finding, Step by Step

In this section, we explain the circuit finding procedure step by step, with additional diagrams to aid comprehension. We start, as indicated previously, by patching direct connections to the logits (Figure 15). This reveals connections to the logits from MLPs 8-11, as well as a9.h1. We continue with the next furthest downstream MLP, MLP 11, and see which nodes influence the circuit via it. Note that the only path through which a node $C$ can influence the circuit via MLP 11 is $(C, \text{MLP } 11, \text{logits})$, in red (Figure 16, right). The results (Figure 16, left) indicate that the other MLPs influence the circuit most through MLP 11.

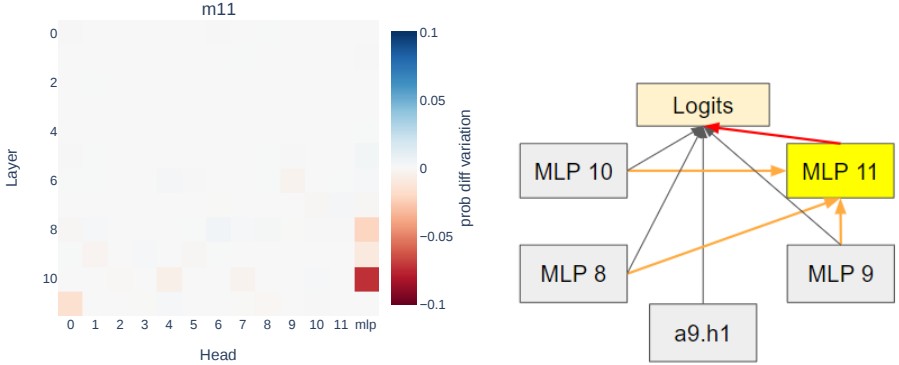

Figure 16: Path Patching Step 2: MLP 11

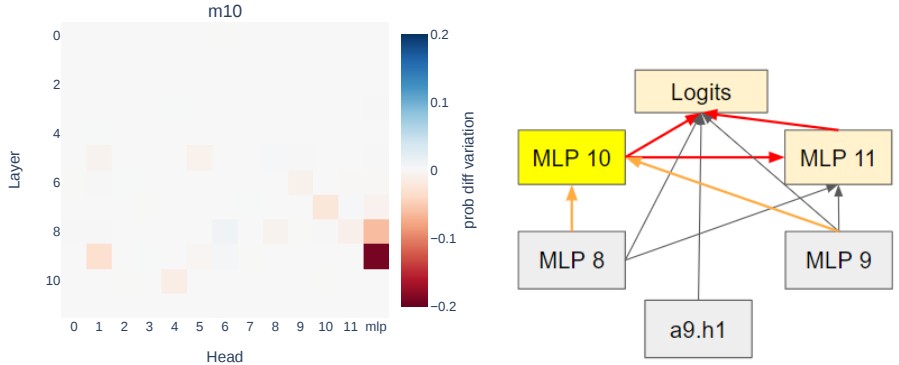

Figure 17: Path Patching Step 3: MLP 10

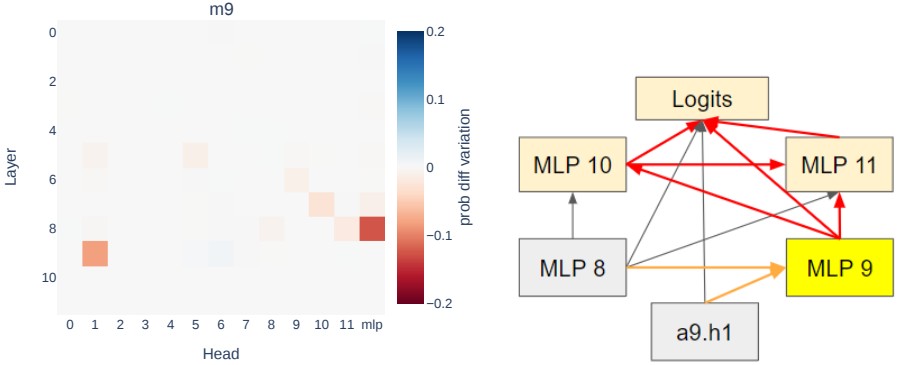

Figure 18: Path Patching Step 4: MLP 9

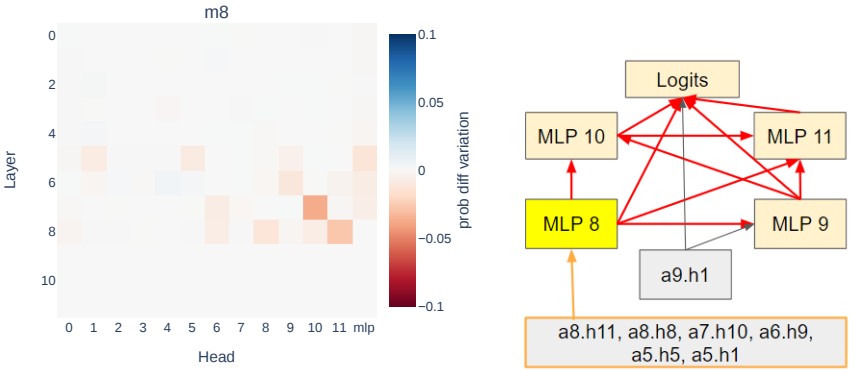

Figure 19: Path Patching Step 5: MLP 8

We continue onward to MLP 10, and see which nodes are most influential on the circuit via it. We consider all the ways in which nodes could contribute via MLP 10 to the circuit, shown in red on the right of Figure 17. The results (Figure 17) again that MLP 10 relies mostly on MLPs 8 and 9. We proceed similarly with MLP 9, considering all the ways in which it influences the circuit; the results in Figure 18 indicate that it relies mostly on MLP 8 and a9.h1.

At this juncture, it might be appropriate to ask which nodes in the graph most influence the circuit via a9.h1. However, we skip this node because, as we explain in the circuit semantics section (Section 3.3), the attention heads are acting together separately from the MLPs, performing different roles. Moreover, when analyzing attention heads, we must consider what influences their queries, keys, and values separately, a complicated task best avoided in the MLP-centric analysis. In Appendix B, we explore in greater detail the nodes that contribute to such attention heads.

Instead, we complete our circuit-finding section by finding nodes that contribute to the circuit via MLP 8. This reveals (Figure 19) the heads that identify YY, completing our initial circuit investigations.

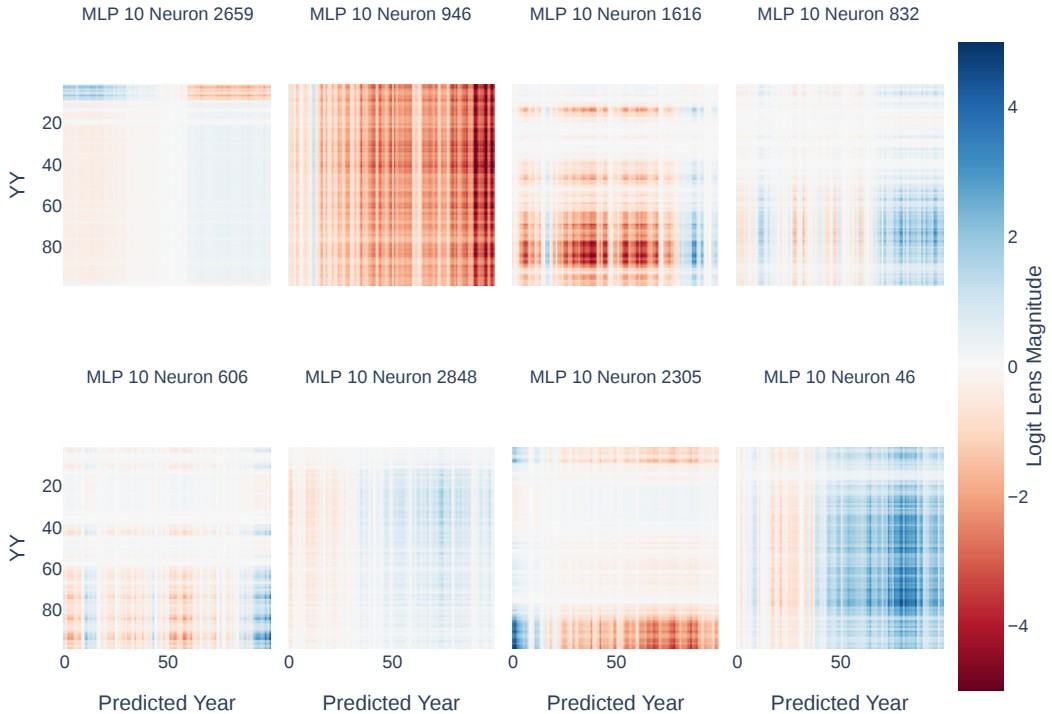

Figure 20: Neuron contributions for each MLP 10 neuron in the top 4-11. Neurons ordered by importance, left-to-right, top-to-bottom. Blue indicates that the neuron upweights a certain predicted year, given a starting year YY, while red indicates downweighting.

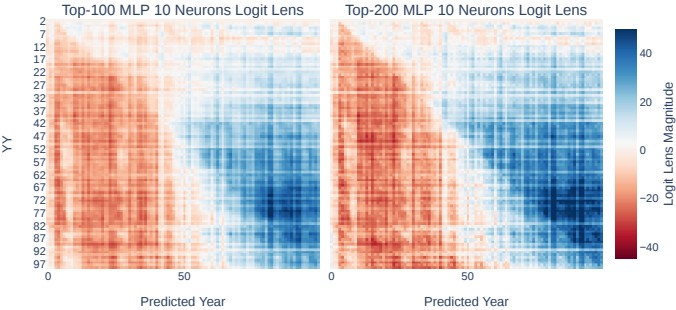

Figure 21: Neuron contributions of the top-100 (left) and 200 (right) neurons in MLP 10. Blue indicates that the neuron upweights a certain predicted year, given a starting year YY, while red indicates downweighting.

## D    MLP 10 Neuron Contributions

In this section, we display the contributions of the top 10 most important neurons of MLP 10, found in Figure 20. Many neurons' contributions are relatively constant across YY; e.g. the 4th most important neuron always upweights later years. Others differ across YY but not predicted year; the 10th most important neuron downweights all years for the last 10 years or so, where correct answers are very few generally. The 3rd most important neuron varies in both dimensions, having 0 contribution for years YY from around 10 to 50, but a distinct pattern for all other YY. Only the first few neurons are very intense in color, as we have fixed the range of the color scale: these neurons are the most important because they cause the greatest changes when they are patched.

Combining these contributions rapidly produces patterns resembling those of the MLP as a whole. We see this weakly by viewing the top-10 neurons' contributions, but more strongly in the top-100 or 200 (of 3072) neurons (Figure 21). In these logit lens diagrams, there is a consistent increase in logit lens magnitude between YY and YY+1, for a given start year YY.

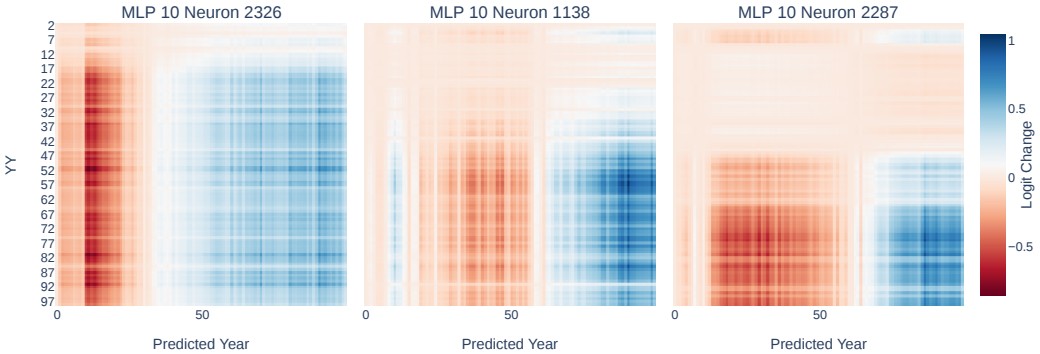

Figure 22: Direct effects of top-3 MLP 10 Neurons

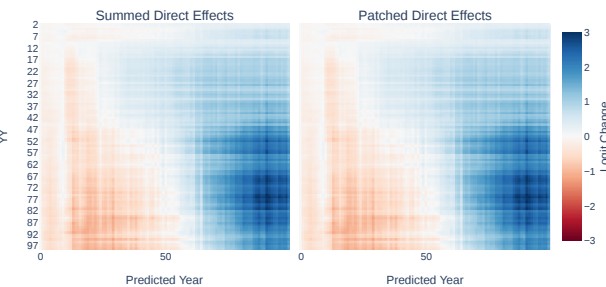

Figure 23: Summed and patched direct effects of top-10 MLP 10 Neurons

# E   Logit Lens vs. Direct Effects via Path Patching

We use the logit lens throughout this paper for consistency, both internally and with prior work. However, the logit lens has flaws: in reality, the residual stream is normalized using layer normalization prior to being transformed into the logits. This nonlinearity could in theory cause the scaling of logit lens to be misleading, perhaps in such a way that affects our conclusions. In this section, we will show that the logit lens results can be recreated with another technique, direct effects via path patching, that avoids this flaw. The two techniques fortunately yield the same insights.

Using the logit lens, we would normally multiply our component of interest's output by the unembedding matrix. To measure direct effects, we take the unpatched, original model's year logits, over each YY, as a baseline. We then patch the direct path between our component and the logits with the 01-dataset, and again record the logits. The difference between the unpatched logits, and the logits when we patch (ablate) our component of interest, reveals the direct effect that said component had. This approach eliminates one major concern of the logit lens: it yields the difference of two sets of logits, which were both produced with layer normalization, and which has interpretable units.

We can perform our neuron-level logit lens experiments by instead patching the direct paths to the top-3 neurons of MLP 10 (Figure 22). The results are essentially identical to the logit lens results, though they differ in magnitude. We can also sum these direct effects, as we summed the logit lens outputs; again, results differ from those of the logit lens only in magnitude (Figure 23). Finally, we can also patch all top-10 neurons as a group, and view their direct effects; these results are identical to those of the summed direct effects (Figure 23). This suggests that our summed logit lens approach genuinely reflected the direct effects that these neurons have on the logits.

We conclude that the results given by the logit lens and those given by direct effects are largely similar, so concerns about the logit lens are not dire. However, we note that all of our logit lens results (including those for entire MLPs and attention heads) are reproducible using direct effects.

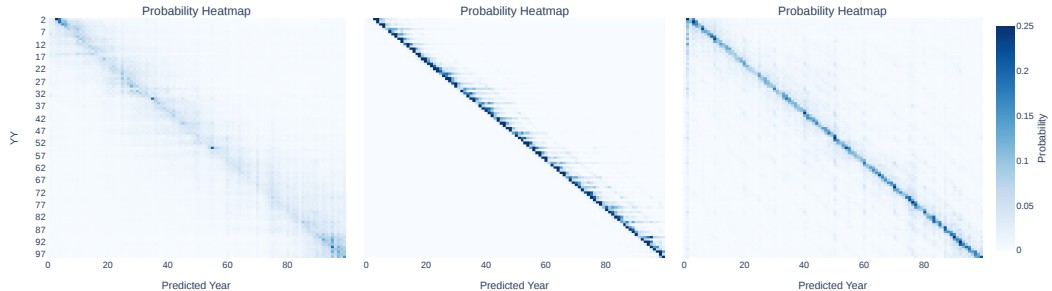

Figure 24: Probability heatmaps for (left to right) "1799, 1753, 1733, 1701, 16YY, 16", "1695, 1697, 1699, 1701, 1703, 17", and "17YY is smaller than 17".

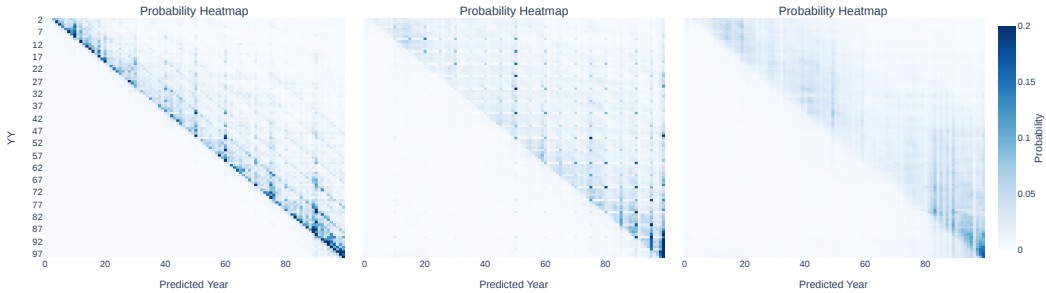

Figure 25: Probability heatmaps for (left to right) "The <noun> started in the year 17YY and ended in the year 17", "The <noun> happened in 17YY. Some years later, it is now the year 17", and "1599, 1607, 1633, 1679, 17YY, 17".

## F  Year-Span Circuit Generalization

In this section we provide evidence for the generalization of the year-span circuit to some (but not all) tasks. For tasks where the model failed entirely, we provide the probability heatmaps (to show its failure). For tasks where the model succeeded, or incorrectly generalized the greater-than circuit, we additionally provide path patching results and logit lens heatmaps to show how circuit structure and semantics are preserved.

**Tasks Failed**  Figure 24 displays probability heatmaps for "1799, 1753, 1733, 1701, 16YY, 16", "1695, 1697, 1699, 1701, 1703, 17", and "17YY is smaller than 17". In the first case, GPT-2 predicts a roughly uniform distribution around YY. In the second case, the right answer varies: though the penultimate number in the provided sequence is YY="03", and the sequence increases by 2 step, depending on YY, we must vary the increase per step, in order to avoid the single-token number "1700". In any case, the model fails to predict the correct answer, often predicting YY+1, although the step is always $> 1$. In the final case, the model always outputs YY.

**Tasks Completed Correctly**  Figure 25 displays probability heatmaps for "The <noun> started in the year 17YY and ended in the year 17", "The price of that <luxury good> ranges from $ 17YY to $ 17", and "1599, 1607, 1633, 1679, 17YY, 17". All tasks are completed successfully, just like year-span prediction, though note that the second task is completed less well, as is visible in its heatmap. Its probability difference is only 75%, as opposed to 90%.

Given this, we proceed using iterative path patching direct logits connections as before; Figure 26 shows the results. All of these plots look almost identical to our original plots, so we evaluate the circuit on each of tasks using the methodology from Section 3.2. This works for the first two tasks, with performance recoveries $> 90\%$, but not the last.

For the last task, we observe that MLP 8 relies also on MLP 7, which in turn relies on two extra attention heads not observed in our original circuit: a7.h11 and a6.h1 (Figure 27). Accounting for this in our circuit leads us back to $> 90\%$ loss recovery.

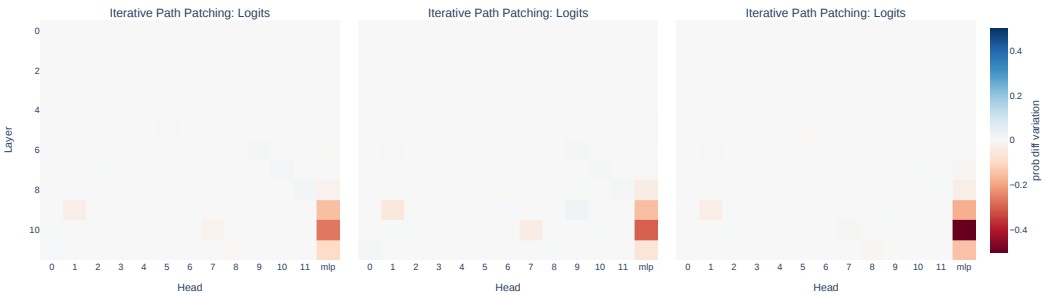

Figure 26: Iterative path patching plots ($C$, logits) for (left to right) "The `<noun>` started in the year 17YY and ended in the year 17", "The price of that `<luxury good>` ranges from $ 17YY to $ 17", and "1599, 1607, 1633, 1679, 17YY, 17".

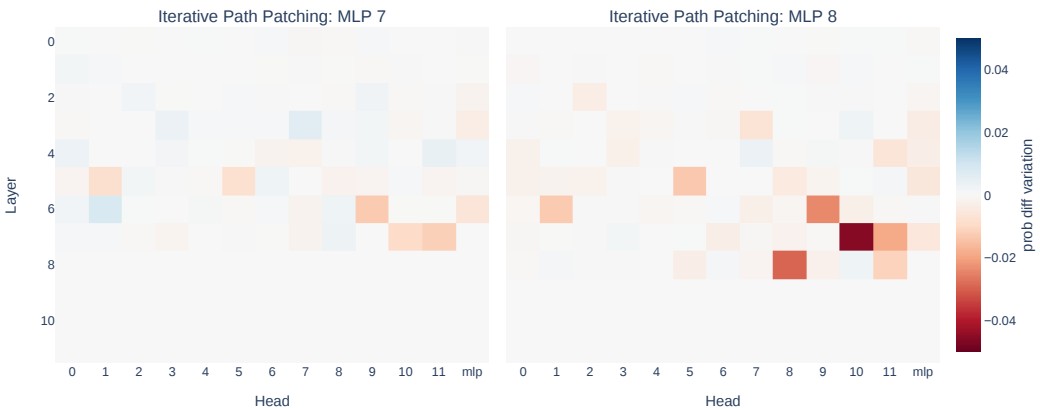

Figure 27: Iterative path patching plots for "1599, 1607, 1633, 1679, 17YY, 17", searching for components that influence the circuit via MLPs 8 (left) and 7 (right).

**Tasks Completed Incorrectly**    Finally, we address the tasks "The `<noun>` ended in the year 17YY and started in the year 17" and "The `<noun>` lasted from the year 7YY BC to the year 7", which do use our circuit, but should not do so.

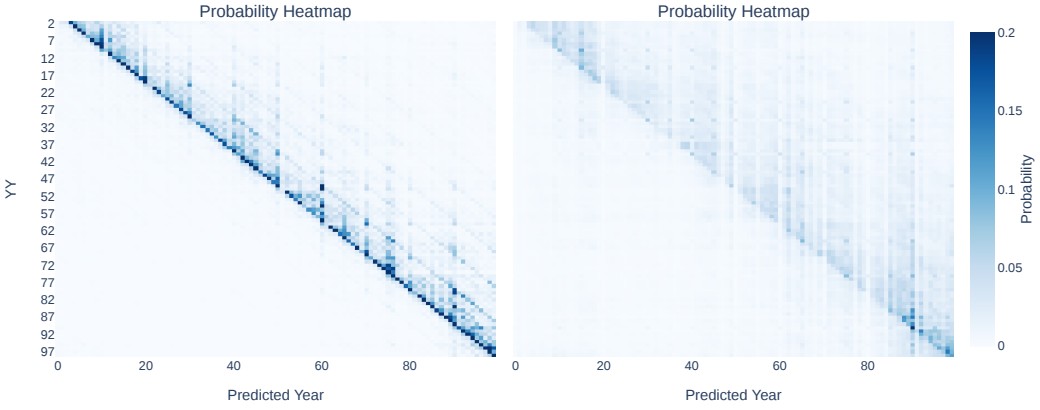

Figure 28: Probability heatmaps for "The `<noun>` ended in the year 17YY and started in the year 17" (left) and "The `<noun>` lasted from the year 7YY BC to the year 7" (right).

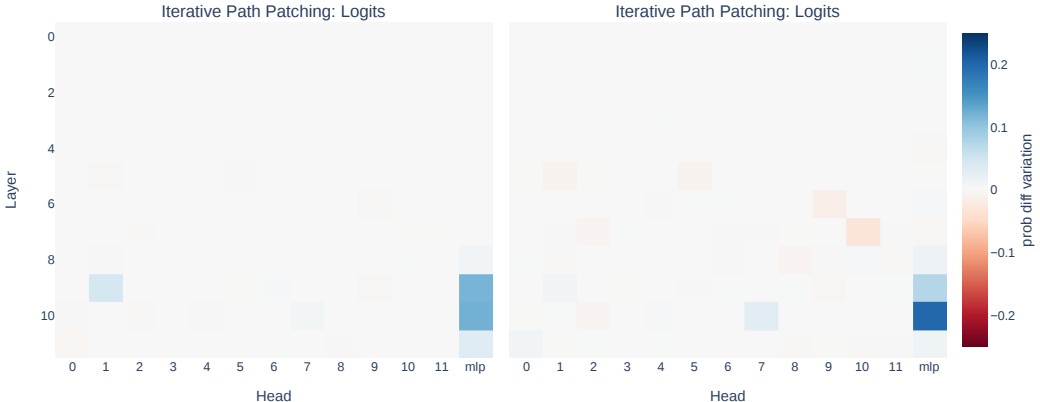

Figure 29: Iterative path patching plots ($C$, logits) for "The `<noun>` ended in the year 17YY and started in the year 17" (left) and "The `<noun>` lasted from the year 7YY BC to the year 7" (right).

Figure 28 displays probability heatmaps for "The `<noun>` ended in the year 17YY and started in the year 17" and "The `<noun>` lasted from the year 7YY BC to the year 7". Both tasks are completed successfully, though note that the latter task is completed less well.

As with the other tasks using this circuit, the iterative path patching plots (Figure 29) look similar to those of year-span prediction. Note that since the goal for these tasks is to produce "less-than", patching and impeding circuit components improves task performance (indicated in blue rather than red), since the circuit performs "greater-than". We evaluate the circuit on each of tasks using the methodology from Section 3.2. This works for both tasks, with performance recoveries $> 90\%$.

## G Noun Pool for Templated Sentences

We use the following nouns in our main template: abduction, accord, affair, agreement, appraisal, assaults, assessment, attack, attempts, campaign, captivity, case, challenge, chaos, clash, collaboration, coma, competition, confrontation, consequence, conspiracy, construction, consultation, contact, contract, convention, cooperation, custody, deal, decline, decrease, demonstrations, development, disagreement, disorder, dispute, domination, dynasty, effect, effort, employment, endeavor, engagement, epidemic, evaluation, exchange, existence, expansion, expedition, experiments, fall, fame, flights, friendship, growth, hardship, hostility, illness, impact, imprisonment, improvement, incarceration, increase, insurgency, invasion, investigation, journey, kingdom, marriage, modernization, negotiation, notoriety, obstruction, operation, order, outbreak, outcome, overhaul, patrols, pilgrimage, plague, plan, practice, process, program, progress, project, pursuit, quest, raids, reforms, reign, relationship, retaliation, riot, rise, rivalry, romance, rule, sanctions, shift, siege, slump, stature, stint, strikes, study, test, testing, tests, therapy, tour, tradition, treaty, trial, trip, unemployment, voyage, warfare, work.

For the `<luxury good>` noun considered in the generalization section, we use a smaller pool of nouns: gem, necklace, watch, ring, suitcase, scarf, suit, shirt, sweater, dress, fridge, TV, bed, bike, lamp, table, chair, painting, sculpture, plant.

## H Applying the Circuits Approach to Other Problems

In order to find a circuit for a given model ability, one must define a task, dataset, and metric; one can then apply the path patching approaches we use. However, task, dataset, and metric design are important; not just any task, dataset, or metric will be compatible. In this section, we discuss the qualities that a task, dataset, and metric should have, in order to be used with our approach.

**Task**  The path-patching approach is compatible with a variety of tasks. The chosen task should:

- Have a clearly delimited set of correct and incorrect answers for each example.

- Require only one forward pass of the model (as opposed to e.g. generation tasks which require multiple passes).
- Be solvable by your model: if your model cannot solve the task, there may be no circuit. At minimum, the model should exhibit consistent behavior (even if it's not exactly correct)

The granularity of insights will depend on the granularity of the task chosen. Complex tasks like natural language inference could require different (sub-)circuits depending on the specific question; it might be hard to find one precise circuit responsible for the task. Smaller, simpler tasks will likely yield easier to interpret results.

For the purpose of this example, we consider the task of fact retrieval, much akin to the task of Meng et al. [28]. Each input will have an (ideally single-token) correct answer, which can be predicted with one forward pass. Moreover, it seems possible that facts are mostly stored and retrieved using the same circuit.

**Dataset**   Path-patching requires two datasets: a normal and corrupted dataset. The normal dataset is just a collection of examples/inputs for the task; its examples should:

- Clearly indicate the task at hand. LMs perform language modeling, and do not natively perform other tasks; they may leak probability to answers that are not correct or incorrect, but simply task-irrelevant. The inputs to the LM should push as much probability as possible onto the task's output space.
- Allow evaluation based on only the distribution over possible next tokens (generated via one forward pass)
- Be representative of your task. Your choice of datasets effectively define the scope of the phenomenon you study, so it is essential that the scopes of the dataset and the intended task match!

Each example from the normal dataset should have a corresponding corrupted example / input. The corrupted input should:

- form a minimal pair with the normal input: they should differ minimally from each other (being the same length, and differing by only one or two tokens)
- elicit a different model response, with a distinct correct answer, compared to the normal input
- belong to the same sort of task. We locate the circuit by activating the same circuit with two different inputs.

For fact retrieval, a normal input could be "Paris is the capital of"; the corrupted counterpart could be "Rome is the capital of". Both of these examples are reasonable input for fact retrieval, but the two will elicit very different responses. Note that an input like "Paris is in" would be less appropriate, because it doesn't clearly indicate that the task is fact retrieval, or what fact should be retrieved.

**Metric**   The metric is a function that takes in model logits and labels. It should:

- Output a real number measuring model behavior/performance on the task.
- Detect small changes in whether the model is behaving according to the normal or corrupted input, or somewhere in between. A continuous loss is thus preferable to metrics like 0-1 loss/accuracy.

One family of metrics used in previous work is the probability assigned to the correct answer(s), minus the probability of the incorrect answer(s) induced by the corrupted input. In the greater-than case, this is $p(y >$YY$) - p(y \leq$YY$)$. For fact retrieval, we would compute $p($France$) - p($Italy$)$. This family of metrics the model is implicitly sensitive to the model generating off-task output as this generally takes away from the probability of correct answers. It is explicitly sensitive to the probability of the corrupted input's answer.

Other metrics are possible. For example, if it is difficult to quantify task performance, but still possible to create minimal pairs, one could simply measure the KL-divergence between the original

and (partially) patched / corrupted distributions. However, this is harder to interpret, and less targeted at your actual task of interest.

## I   Computational Resources

All experiments were performed on an Nvidia A100 GPU. The path patching experiments and circuit semantics experiments take no longer than an hour to complete. The generalization experiments take a similar amount of time, being very similar to the original experiments. The neuron-level experiments (in particular finding top neurons) can take multiple hours to run. Overall, the final experiments can be run in less than 24 hours.

