# OpenReview forum: "How does GPT-2 compute greater-than?: Interpreting mathematical abilities in a pre-trained language model"
_NeurIPS.cc/2023/Conference — NeurIPS 2023 poster_

### Official Review · Reviewer_WeSY · 2023-07-08

**Soundness:** 4 excellent
**Presentation:** 3 good
**Contribution:** 4 excellent
**Rating:** 6
**Confidence:** 4

**Summary:**

The paper attempts to improve our understanding on the source of a certain ability (the use of "greater than" in mathematical tasks) in LMs. To this end, the authors outlined a circuit in GPT-2 with interpretable structure and semantics, adding to the evidence that circuits are a useful way of understanding pre-trained LMs.

Compared to existing work, the use of circuits is more fine-grained, and reveals special insights of such ability. The experiments are not only conducted on math tasks, but also extended to adjacent tasks that are also involved with "greater than" relations.

Although there might be some concerns on whether these findings can be generalized to ever larger models, the reviewer still thinks it is a good paper as it shows a unique way to understand deep neural networks.

**Strengths:**

1. Novel approach on understanding the ability of LMs.
2. Extensive experiments and clear presentation.

**Weaknesses:**

1. The conclusion might be limited to certain model type (auto-regressive LMs such as GPT series).
2. Relatively narrow choice of tasks (only studying the "greater than" relationship).

**Questions:**

Have you ever conducted experiments on "smaller than"? I'm curious about whether the findings are the mirrored version of "greater than", or they are total different from each other. If we can find some "semantic relationship" among the circuits, that will make such pipeline easier to be generalized.

**Limitations:**

As mentioned in the Weakness section.

---

> ### Author Rebuttal · Authors · 2023-08-08
>
> Thank you for your review! Here are our responses:
> > The conclusion might be limited to certain model type (auto-regressive LMs such as GPT series).
>
> We agree that some of our conclusions, such as our specific circuit, are limited. Other findings, such as the way attention heads and MLPs work together to solve problems, or the form of partial generalization we observe, could surface in other models as well. For more discussion, see the **Scaling** section of the global response. More broadly, our methodology—from circuit finding to analysis—can easily be applied to other autoregressive or masked transformer LMs.
> > Relatively narrow choice of tasks (only studying the "greater than" relationship).
>
> This is true. We choose our task in part because GPT-2 small is not very mathematically competent: its performance on other math tasks is generally not high enough to be worth interpreting. However, we find that a small problem scope also allowed us to dig deeper and find a detailed circuit; see the **Problem Scope** section of the global response for more details.
> > Have you ever conducted experiments on "smaller than"? I'm curious about whether the findings are the mirrored version of "greater than", or they are total different from each other. If we can find some "semantic relationship" among the circuits, that will make such pipeline easier to be generalized.
>
> Yes, we have conducted experiments on less-than! These experiments are in the generalization section (Sec. 5) of our paper. To summarize, given prompts like “17YY is less than 17”, GPT-2 tended to predict YY. Given prompts like “The war ended in 17YY and started in 17”, GPT-2 used the greater-than circuit to predict a year >YY! This is to say that GPT-2’s less-than behavior is totally different from its greater-than behavior: there is no consistent behavior, and thus no real circuit underlying it.
>
> This might seem disappointing—as you say, it would be nice to find a semantic relationship between two circuits that reflects some sort of deeper understanding of mathematics or the order of numbers. However, this is still interesting: it hints that models might perform something in between simple memorization and full generalization. GPT-2 uses our circuit in distinct greater-than scenarios, so there is some partial generalization; however, this does not stem from real mathematical abilities. We think that this partial, incomplete generalization, reflects LMs’ abilities: close to humans’ on the surface, but far in terms of deeper understanding.
>
> Let us know if you have any other questions or suggestions to improve our work!

---

> > ### Comment · Reviewer_WeSY · 2023-08-19
> > **Thank you for the response!**
> >
> > Thank you for the response. I would like to keep my current rating.

---

### Official Review · Reviewer_BwzU · 2023-07-11

**Soundness:** 4 excellent
**Presentation:** 2 fair
**Contribution:** 3 good
**Rating:** 6
**Confidence:** 4

**Summary:**

This work investigates in depth the mechanism of GPT-2 small to compute the "greater than" function. Specifically, the work isolates a portion of the computational units that are causally related to making plausible predictions for inputs similar to: "The war lasted from 1731 to 17__" The isolation of computational units is done via a recent previously proposed method called "path patching". The resulting isolated subnetwork is shown to be necessary and approximately sufficient for making plausible predictions for this task. The authors also show that this subnetwork generalizes to other types of templates that require prediction of a greater number.

**Strengths:**

- In-depth mechanistic analysis of a language model that would be of interest to some of the NeurIPS audience
- Strong scientific validity. I appreciate that the authors thought through possible confounders and did extra experiments to validate their findings

**Weaknesses:**

1. In the current format, the contributions of the current work are difficult to disentangle from previous work. The impact can be strengthened by clearly stating what the current work contributes above previous work, and by a detailed explanation of how a researcher can build on the analysis framework to investigate other emergent skills/properties of language models.

2. The work investigates only one model - GPT-2 small. Investigating additional models can help understand how these mechanisms emerge, and how they are influenced by scale and amount of training data.



**Questions:**

Major:

Q1. Related to weakness 2: Which part of the interpretability techniques are borrowed from previous work, and what is newly proposed here? How would a researcher interested in investigating a different function or a different model do this using the proposed framework?

Q2. For the sufficiency and necessity experiments in Section 3.2: if the identified circuit is sufficient and necessary, then does it matter what input is provided to nodes outside of the circuit? How would providing random input to the nodes outside the circuit change the results, while providing the original dataset to the nodes inside the circuit?

Q3. Which of the results do you expect to hold in larger GPT-2 variants, and how can the proposed analysis framework be used to investigate the emergence of properties at different scales of models / training data?


Minor:

- Fig 3 is difficult to understand on its own. The change in the direct connection between MLP 10 and the logits is difficult to spot between A and B. I recommend leaving the direct connection where it is in A but coloring it in a different color, and writing a more descriptive caption. Also the caption in C says that MLP11 receives input 01 but there are no direct or indirect arrows between the 01 input and MLP11.

- Also it’s not quite clear what it means to “patch” a path. How does one ensure that the perturbed input that is provided to the patched MLP 10 does not influence all downstream components of MLP 10?

- The use of footnotes is standard for NLP venues but not so much for ML venues. I personally find footnotes distracting.

- L178: typo: “and and”


**Limitations:**

See Weaknesses

---

> ### Author Rebuttal · Authors · 2023-08-08
>
> Thank you for your careful review! We’ve answered your questions below, omitting the weaknesses, as they correspond roughly to Q1 and Q3. Regarding our contributions: please see the **Contributions** section of the general response, and Q1. We will revise our paper to clarify our contributions, and add details regarding how to apply our methods to other models.
> > Q1. Related to weakness 2: Which part of the interpretability techniques are borrowed from previous work, and what is newly proposed here? How would a researcher interested in investigating a different function or a different model do this using the proposed framework?
>
> The **Methodological Contributions** section of the global response tackles this contributions aspect of this question.
>
> Our methods are quite general and can be applied to other tasks/models. Generally, one needs to define a task that the model can perform, with a corresponding dataset and metric. One also defines a corrupted (patch) dataset that induces different model behavior, measurable by the metric. Then, one can corrupt connections using the patch dataset, and determine which connections are important.
> We suggest patching specific connections, rather than patching the output from one node to all other nodes, in order to gain a finer-grained understanding of how tasks are performed. Our work also shows the potential of testing the found circuit on new datasets for similar tasks. Note that doing this research on new models can involve some model reimplementation, in order to enable manipulation of the computational graph, or allow intervention on its intermediate values.
> > Q2. For the sufficiency and necessity experiments in Section 3.2: if the identified circuit is sufficient and necessary, then does it matter what input is provided to nodes outside of the circuit? How would providing random input to the nodes outside the circuit change the results, while providing the original dataset to the nodes inside the circuit?
>
> This is an interesting question! Our circuit contains all nodes that compute greater-than, given that the quantity being computed is greater-than. That is, if the context requires greater-than output, our circuit computes it. Thus, when we ablate our model, we give the ablated portion input tokens that still require greater-than output, while altering the quantity YY that the output should be greater than.
>
> But if the input to the ablated nodes suggests that some other sort of output is needed, behavior could differ. Our circuit might not function, or another circuit, in the ablated portion of the model, triggered by the different input, could dominate model behavior. And, if the other nodes’ values were set to something totally out of distribution, it’s tough to know precisely what the model’s output would look like. We think that your question is deeply tied to questions like “What contexts cause this circuit to activate, and how do LMs detect such contexts?” which should be studied in future work.
> > Q3. Which of the results do you expect to hold in larger GPT-2 variants, and how can the proposed analysis framework be used to investigate the emergence of properties at different scales of models / training data?
>
> The **Scaling** section of the global response answers this question. In essence, the mechanistic motifs that we find—the co-operation of attention heads and MLPs, as well as partial circuit generalization—seem likely to generalize to larger models.
>
> As for our methods, the circuits framework scales well on a technical level, especially when automated. It is quite general: if you can define a task with a normal and a patched dataset, as well as a metric, you can apply circuit-finding techniques. The more challenging question is how to assign semantics to these circuits at scale. Others have tried to automate e.g. using other models to assign semantics to model neurons, but this sort of technique is still immature. However, we think that automatically assigning semantics to model internals is challenging for most interpretability frameworks, not just ours.
> > Fig 3 is difficult to understand on its own[...]
>
> Good suggestion; we’ll use a different color for the removed connection between MLP10 and the logits. And indeed, the caption of part C is wrong: MLP 11 does not receive 01-input. We’ll make both of these changes—thanks for your careful attention to our diagrams.
> > Also it’s not quite clear what it means to “patch” a path. How does one ensure that the perturbed input that is provided to the patched MLP 10 does not influence all downstream components of MLP 10?
>
> In the framework we use (rust-circuit), we directly manipulate the model’s computational graph. So, to patch the direct MLP10->logits path, we can copy MLP10 and its ancestor nodes, and corrupt the token inputs to that copy. We then replace the original MLP10->logits edge with a corrupted MLP10->logits edge; we don’t touch the edges between the original MLP10 and other nodes. Thus the corrupted MLP10 is an ancestor of only the logits; any other components with MLP10 as an ancestor see the original MLP10.
>
> Simple path-patching can be done using vanilla PyTorch models / hooks, or via a package like TransformerLens—let us know if you’d like details! For a more formal exploration of path patching, see [2] in the global response.
>
> Please let us know if you have any other questions or ideas regarding how to improve our work!

---

> > ### Comment · Reviewer_BwzU · 2023-08-14
> >
> > Thanks for the response. The added clarifications re path patching and the contributions are helpful. To me the major limitation to how impactful this work will be remains the ease with which other researchers will be able to build on the interpretability approach proposed here. I urge the authors to include a more specific discussion of this in a future revision. Specifically statements like "one needs to define a task that the model can perform, with a corresponding dataset and metric." are not sufficient, and the characteristics of the desired datasets and metrics, with respect to the task, need to be discussed. That said, I do believe in the importance of the work, even if it ends up being a one-off scientific investigation.

---

> > > ### Author Response · Authors · 2023-08-16
> > >
> > > Thanks for your response! We’re glad to hear that the clarifications were helpful, and that you believe in the importance of our work.
> > >
> > > We also value enabling others to build on our work. As part of this, we will release our code, included in the supplementary material. As suggested, we will also add details about how to choose tasks, metrics, and datasets, in order to aid future researchers. While we can’t upload revisions to our paper, we’re happy to share a rough outline of this information here.
> > >
> > > Let us know if you’ve got any other questions, and if there’s anything we can do to help you raise your score!
> > >
> > > ---
> > >
> > > ### Task
> > > Our path-patching approach is compatible with a variety of tasks. The chosen task should:
> > > 1. Have a clearly delimited set of correct and incorrect answers for each example.
> > > 2. Require only one forward pass of the model (as opposed to e.g. generation tasks which require multiple passes).
> > > 3. Be solvable by your model: if your model cannot solve the task, there may be no circuit. At minimum, the model should exhibit consistent behavior (even if it’s not exactly correct)
> > >
> > > Keep in mind that the granularity of insights will depend on the granularity of the task chosen. Complex tasks like natural language inference could require different (sub-)circuits depending on the specific question; it might be hard to find one precise circuit responsible for the task. Smaller, simpler tasks will likely yield easier to interpret results.
> > >
> > > For the purpose of this example, we’ll consider the task of fact retrieval. Each input will have an (ideally single-token) correct answer, which can be predicted with one forward pass. Moreover, it seems possible that facts are mostly stored / retrieved using the same circuit.
> > >
> > > ### Dataset
> > > Path-patching requires two datasets: a normal and corrupted dataset. The normal dataset is just a collection of examples/inputs for the task; its examples should:
> > > 1. Clearly indicate the task at hand. LMs perform language modeling, and do not natively perform other tasks; they may leak probability to answers that are not correct or incorrect, but simply task-irrelevant. Your inputs should push as much probability as possible onto the task's output space.
> > > 2. Allow evaluation based on only the distribution over possible next tokens (generated via one forward pass)
> > > 3. Be representative of your task. Your choice of datasets effectively define the scope of the phenomenon you study—make sure the scopes of your datasets and your intended task match!
> > >
> > > Each example from the normal dataset should have a corresponding corrupted example / input. The corrupted input should:
> > > 1. form a minimal pair with the normal input: they should differ minimally from each other (being the same length, and differing by only one or two tokens)
> > > 2. elicit a different model response, with a distinct correct answer, compared to the normal input
> > > 3. belong to the same sort of task. Remember that we locate the circuit by activating the same circuit with two different inputs.
> > >
> > > For fact retrieval, a normal input could be "Paris is the capital of"; the corrupted counterpart could be "Rome is the capital of". Both of these examples are reasonable input for fact retrieval, but the two will elicit very different responses. Note that an input like "Paris is in" would be less appropriate, because it doesn't clearly indicate that the task is fact retrieval, or what fact should be retrieved.
> > >
> > >
> > > ### Metric
> > > The metric is a function that takes in model logits and labels. It should:
> > > 1. Output a real number measuring model behavior / performance on the task.
> > > 2. Detect small changes in whether the model is behaving according to the normal or corrupted input, or somewhere in between. A continuous loss is thus preferable to metrics like 0-1 loss/accuracy.
> > >
> > > One family of metrics used in previous work is the probability assigned to the correct answer(s), minus the probability of the incorrect answer(s) induced by the corrupted input. In the greater-than case, this is p(y>YY) - p(y<=YY). For fact retrieval, we would compute p(France) - p(Italy). This family of metrics the model is implicitly sensitive to the model generating off-task output as this generally takes away from the probability of correct answers. It is explicitly sensitive to the probability of the corrupted input’s answer.
> > >
> > > Other metrics are possible. For example, if it is difficult to quantify task performance, but still possible to create minimal pairs, one could simply measure the KL-divergence between the original and (partially) patched / corrupted distributions. However, this is harder to interpret, and less targeted at your actual task of interest.

---

### Official Review · Reviewer_GMC8 · 2023-07-17

**Soundness:** 3 good
**Presentation:** 1 poor
**Contribution:** 2 fair
**Rating:** 4
**Confidence:** 5

**Summary:**

This paper explores how GPT-2 performs the ``greater-than'' operation by analyzing its circuit. The authors construct a template of the operation and define two scores to evaluate the performance of GPT-2. They first evaluate that the found circuit is indeed important for performing the greater-than operation, and then find that top MLP layers in GPT-2 compute the operation.

---
Rebuttal response:

Thanks for the detailed explanations. Some of my concerns about the details are addressed. Thus, I would like to raise the soundness score. Regarding the contribution, I agree that the method part under the mechanistic interpretability scenario is sufficient. However, the findings are not attractive to me. Thus, I will raise the score to 2. Regardless of whether the paper is accepted or not, I would like to suggest that the presentation could be improved.

**Strengths:**

This paper explores a very interesting question for LM understanding: how LMs perform tasks inside its architecture. The analysis method, i.e., looking at the circuit of LM, is suitable for this kind of interpretation.

**Weaknesses:**

1. The key contribution is not clear. This paper uses the existing analysis method for interpretation. But, to research the question (how LM performs math calculation), only a simple template and a toy task (greater-than operation) are discussed for GPT-2. There is a limited contribution to the analysis.
Besides, for findings, such as GPT-2 performs the operation on top MLP layers, many existing works (e.g., probing works) have also found that, and existing findings are even more comprehensive.
Thus, the contribution of this work is not clear. I would suggest that the authors should focus on one side (e.g., analysis method, interpretation task/data/template design) and dig deeper.


2. The presentation needs to be improved. The essential information should be clearly introduced and technical details could be moved to the appendix.
For example, in section 2, it says that only single-token numbers are considered in this work (lines 65-72). We know that it is because of GPT-2's BPE tokenizer. There is no need to give specific examples for that. But for the dataset introduction, what's the size of it, how did you construct it (and why in that way), these parts are not well introduced.
Besides, it would be better to use some tables in section 3 to make it more readable.

3. Generalization. As mentioned above, generalizing the analysis method to large LMs with more complex math problems is not intuitive. With different tokenizers (e.g., LLaMA tokenizer), the analysis template in this paper would fail. Besides, recent instruction-tuned LLMs may have very different conclusions compared to GPT-2.

**Questions:**

Why the curve in the right figure of Figure 2 are continuous? Some tokens (e.g., 00) are not considered as described in the paper.

(Line 81): Why mention ``GPT-2's training data''? Do you finetune GPT-2 or just use it in the zero-shot setting?

(Line 87): This should be average instead of sum, right? Otherwise, it would not range from -1 to 1.

(Line 305): Typo: there are double ``from''.

**Limitations:**

Yes

---

> ### Author Rebuttal · Authors · 2023-08-08
>
> Thank you for your critique! We hope to resolve your concerns by clarifying both the intent of our work and our contributions.
> > The key contribution is not clear. …There is a limited contribution to the analysis.
>
> We clarify our key contributions and analyses in the global response’s **Contributions** section.
> > This paper uses the existing analysis method for interpretation.
>
> This isn’t quite correct. We introduce new analyses within the circuits framework: we patch multi-node paths, and test our circuit in scenarios distinct from the one in which we found it. Even when using existing methods, we don’t rely on a set of established tools for circuits analysis: the circuits literature is young, and no such toolkit exists. One goal of this paper is to set a methodological standard for circuits research.
> > But, to research the question (how LM performs math calculation), only a simple template and a toy task (greater-than operation) are discussed for GPT-2.
>
> The greater-than task is not intended to explain how LMs perform all math, but rather to examine one small part of LMs’ mathematical abilities in detail. In fact, our results suggest that the greater-than circuit may not be part of a larger set of math capabilities; instead it is a mechanism between memorization and generalization. By studying a small task, we found an interesting mechanism with broader implications for math and generalization in LMs.
> > Besides, for findings, such as GPT-2 performs the operation on top MLP layers, many existing works (e.g., probing works) have also found that.
>
> We respectfully disagree; our findings are not a repetition of existing work. Our core finding is a precise and novel circuit for a math operation in a pre-trained LM—a contribution beyond superficial, layer-wise analyses of top MLP layers. Our circuit generalization results are also absent from prior work. See the **Contributions** section of the global response for more details.
>
> We stress that even simple conclusions like “top MLP layers do a lot of work” are not obvious or settled, but rather up for debate. While existing work has posited a role of certain (not top) MLPs in e.g. factual associations [1], other work disagrees, localizing this elsewhere [2].
>
> Our methodology is meaningfully distinct from non-causal methods like probing. Because it is not causal, probing often finds information in model representations that is not actually used by models [3]; our methods avoid this pitfall. Thus, even if probing work has found similar results, reproducing these with more trustworthy causal methods is still valuable.
>
> That said, we are happy to discuss more related work in the paper; feel free to highlight such work, especially if you feel it overlaps with ours.
> > The presentation needs to be improved. The essential information should be clearly introduced and technical details could be moved to the appendix…the dataset…[is] not well introduced. Besides, it would be better to use some tables in section 3...
>
> Some of the dataset information unintentionally bleeds over from the Task and Dataset section into the following Qualitative Evaluation section. We will fix this, prioritize essential dataset information, and organize some section 3 results in tables.
> >Generalizing the analysis method to large LMs with more complex math problems is not intuitive. With different tokenizers…the analysis template in this paper would fail. Recent instruction-tuned LLMs may have very different conclusions.
>
> We discuss this in the **Scaling** section of the global response. Circuits scale and have been used at larger scales in the time since submission. While our template may not work for all math problems, the challenge of crafting templates and data is part of all interpretability work, and is not unique to our methods.
>
> Regarding findings, we can’t know for sure if large or instruction-tuned LMs process math as GPT-2 does; however, work like ours is a necessary first step towards understanding math in those large models.
> > Why the curve in the right figure of Figure 2 are continuous? Some tokens (e.g., 00) are not considered as described in the paper.
>
> We could have used a scatterplot, as we only have probabilities for discrete years, but the curve had more visual appeal. We do not include 00 as a potential start year; however, the figure’s x-axis is the predicted year y, and the y-axis is p(y|prompt). We do measure p(00|prompt); it stays low, as it is never a correct answer, and is an unnatural tokenization of centuries.
> >(Line 81): Why mention ``GPT-2's training data''? Do you finetune GPT-2 or just use it in the zero-shot setting?
>
> No, we don’t fine-tune GPT-2—we do zero-shot evaluation. We meant to suggest that the durations that GPT-2 predicts for each event are likely related to patterns in its (pre-)training data. However, we can remove this speculation.
> >(Line 87): This should be average instead of sum, right? Otherwise, it would not range from -1 to 1.
>
> No, the sum is correct. The probability difference (PD) for an example with starting year YY is the sum of probability assigned to years > YY minus the sum of the probability assigned to years <= YY. PD is maximized (PD=1) when all probability is assigned to years > YY, and none to any other tokens, including non-year tokens. It's minimized (PD=-1) when all probability is assigned to years <= YY. We aggregate PD by averaging over examples in our dataset.
>
> We hope this answers your questions. Other reviewers found our analyses satisfying, and our findings interesting; let us know how we can improve the paper, and make you feel the same way too!
>
> ### References
> [1]: Meng et al. 2022. Locating and Editing Factual Associations in GPT. NeurIPS
>
> [2]: Hase et al. 2023. Does Localization Inform Editing? Surprising Differences in Causality-Based Localization vs. Knowledge Editing in Language Models. ArXiV
>
> [3]: Belinkov. 2022. Probing Classifiers: Promises, Shortcomings, and Advances. Computational Linguistics

---

> ### Author Response · Authors · 2023-08-16
> **Thank you for your update!**
>
> Thanks for your response—we’re glad to hear that you’re more convinced about the contributions and soundness of our work. We will revise our paper to improve its presentation and clarity as you suggest.

---

### Official Review · Reviewer_eiM2 · 2023-07-19

**Soundness:** 3 good
**Presentation:** 4 excellent
**Contribution:** 2 fair
**Rating:** 7
**Confidence:** 3

**Summary:**

This paper presents an analysis on how the greater-than operator is implemented in the weights of GPT-2 small. They do so by tasking the model with completing sentences of the form "[something, e.g., a war or time period] lasted from $y_1$ to $y_2$" where $y_1, y_2$ are years. The idea is that GPT often assigns values $>y_1$ much higher probability than values $\leq y_1$, when trying to predict $y_2$. This is likely learned due to statistical co-occurrence, but the authors ask _what mechanism_ encodes this behavior in the weights.

**Strengths:**

* The paper is well-written, easy to follow, and well-motivated. The explanations are clear, with limitations stated clearly.
* I enjoyed the exposition of how the circuit was discovered and the interpretation of its semantics.
* It's generally hard to find simple, easy-to-evaluate tasks that GPT-2 small is proficient at, so kudos here.

**Weaknesses:**

Most of my "concerns" are really more aptly formulated as questions. Interpretability is still very nascent, and to me this kind of contribution is meaningful, in spite of (or actually because of) the many unanswered questions it raises.
* I will say that I'm not 100% convinced by the argument at the end of Section 5, that this circuit is not using memorization and actually generalizes. It's hard to know what that means and to show it, especially on a small toy dataset with only four-digit numbers (really, two-digit completions). I think you'll need to demonstrate the same phenomenon on much larger datasets, as it's not super unlikely that the model could memorize the relationship between numbers from 0-99. You even show via ablation that the representational structure discovered by PCA is not a complete explanation. Maybe reframe these claims, or perhaps provide more convincing evidence?
* I think you can _really_ nail the "so what" of the paper by showing how knowledge of this circuit can be used to improve the reliability of the greater-than operation. Do you think you could design a circuit by hand that outperforms the existing one on this task? Or modify the existing one? What would happen if you integrated it into the model? Can you move the circuit to another location? Why is the circuit the way it is?

**Questions:**

* In Section 3.2: what does the PD look like when you give most of the network the 01-dataset but patch the regular dataset into a circuit _other than_ the one you discovered? Based on Figure 4 I'm guessing it'd be lower, but I don't have a great intuition for how much lower.
* In Section 3.3:
  * Figures 6, 7 restrict the plotted tokens to the numbers. In absolute terms, were those numerical tokens promoted the most by the logit lens? Did you find any surprises, i.e., other tokens being ranked higher than numbers in inner product?
  * MLP 8 is interesting! What's going on with it? What does it mean for an MLP to contribute indirectly? What could it be doing? Relatedly, you might have answered this already, but do you have a better intuition for how MLPs 8-11 interact? Whether and why you can't just remove one of them?
* What kinds of analyses did you have to come up with for this paper? I'm not very familiar with circuits; are most of the techniques here standard or did you have to come up with new tricks while, e.g., explaining the semantics of the circuit? How do these tricks generalize to other problems in interpretability?

---

> ### Author Rebuttal · Authors · 2023-08-08
>
> Thanks for your attentive review! We answer your questions below; let us know how else we can improve the paper!
> > I will say that I'm not 100% convinced by the argument at the end of Section 5, that this circuit is not using memorization and actually generalizes. [...]
>
> Thanks for this comment—we’ve been thinking about this post-submission as well. We agree that our original framing suggested generalization too strongly. There is a real possibility that our circuit has memorized greater-than; its limited ability to activate in new contexts could come from the model’s ability to recognize relevant contexts / patterns, as NNs are suggested to do. We will revise the paper to reframe our claims, emphasizing this as an interpretation. We do want to add, though, that we find this possibility very interesting as well, and believe that mechanistic evidence of this sort of memorization is also valuable.
>
> > Using our circuit to improve reliability of greater-than/outperform the existing circuit/etc.
>
> We agree that this sort of contribution would be valuable, and regret that it falls out of the scope of our work. This study mostly answers “When GPT-2 computes greater-than, how does it do so?”. However, the best way to improve GPT-2’s greater-than reliability would be to cause the circuit to activate in situations where it should, but doesn’t currently. To achieve this, it would’ve been better to ask “How does GPT-2 decide if it should compute greater-than?”. If we knew precisely when/why our circuit activated, we could cause it to activate in new scenarios. This would be exciting follow-up work.
>
> There is other work that uses circuits for more practical purposes; for example, other studies have cut relevant edges from model graphs to curtail bad behavior [1]. We hope to investigate such techniques in the future.
>
> [1]: Li et al. 2023. Circuit Breaking: Removing Model Behaviors with Targeted Ablation. ICML Deployable Generative AI Workshop
> > In Section 3.2: what does the PD look like when you give most of the network the 01-dataset but patch the regular dataset into a circuit other than the one you discovered?
>
> Interesting question! Choosing the “other” circuit is tricky, but we tested this, selecting circuits around the same size/location as the original. We found that performance is related to how well input->attention->MLP (especially 9/10)->logits paths are preserved. If the path is interrupted (i.e. no attention head or no MLP overlap with the original circuit), PD is very low (-37%). If at least one path is preserved, PD improves with each additional component in common with the original circuit; MLPs have the biggest impact.
> > Figures 6, 7 restrict the plotted tokens to the numbers. In absolute terms, were those numerical tokens promoted the most by the logit lens? Did you find any surprises, i.e., other tokens being ranked higher than numbers in inner product?
>
> Yes, numerical tokens were the top-ranked tokens for both MLPs and attention heads. For the attention heads, only the top ~3 tokens were numbers, while for the MLPs, the top-k tokens were numbers at higher k. We attribute this to the fact that the MLPs upweight a set of numbers, while the attention heads generally upweight only the one (their top-1 token, generally).
>
> That the numerical tokens are the top tokens is a little surprising—in theory, other tokens could be boosted, and later pushed down by other modules; modules might also be making high-magnitude changes to other words’ logits at the same time as they upweight the right answer. However, other work has also observed that LMs create predictions by promoting the right answer (not downweighting others). This seems like a useful observation to add to the paper.
> > MLP 8 is interesting! What's going on with it? What does it mean for an MLP to contribute indirectly? What could it be doing? Relatedly, you might have answered this already, but do you have a better intuition for how MLPs 8-11 interact? Whether and why you can't just remove one of them?
>
> We also find MLP 8 interesting! Its output, like that of the attention heads, is important to making MLPs 9-11 upweight the correct values; unlike them, though, it doesn’t clearly upweight YY! Its contributions may relate to the logits of non-years—or perhaps logit space isn’t the correct one to view its contributions in. That is, we know that MLP 8 helps the other MLPs, but we’re still unsure of how.
> MLPs 8-11 are interconnected in our circuit. Removing one is possible, but removing e.g. an MLP with strong direct effects like MLP 10 or 11 would drastically reduce the portion of the circuit upweighting the right tokens. Not all connections are equally important though: we consider all MLPs interconnected for simplicity, but for some MLP pairs, the edges between them can be cut without a performance drop.
> >What kinds of analyses did you have to come up with for this paper? [...A]re most of the techniques here standard or did you have to come up with new tricks while, e.g., explaining the semantics of the circuit? How do these tricks generalize to other problems in interpretability?
>
> We address this in the **Methodological Contributions** section of the global response. Many of our semantics techniques are from other work. However, the circuits literature is rather young, with little in the way of standard techniques; with this paper, we aimed to develop a toolkit for studying circuit semantics.
>
> Here are some techniques we used, and how we feel they generalize to new categories of problems:
> - The logit lens characterizes individual nodes; combining it with complex path-patching can tell you what nodes compute, and what nodes use the information computed. These techniques are generally applicable.
> - Neuron-level interventions seem useful for MLP semantics.
> - PCA on the output of individual nodes was visually appealing and interesting, but difficult to transform into concrete, causal insights.

---

> > ### Comment · Reviewer_eiM2 · 2023-08-19
> >
> > Thanks for all the care you put into this response, and the others as well! I'll adjust my score accordingly.

---

### Official Review · Reviewer_Nf2K · 2023-07-28

**Soundness:** 2 fair
**Presentation:** 4 excellent
**Contribution:** 3 good
**Rating:** 6
**Confidence:** 3

**Summary:**

The paper studies and tries to explain how GPT-2 (small) could be computing the mathematical operation of "greater than". A "circuit" (or a subgraph of GPT-2 model's computation graph) is identified by iteratively "patching" individual components to find which components are most responsible for making the correct prediction for this task. The main claim of the paper is that they identify this circuit and show that this circuit is used by GPT-2 small in other contexts that need the greater than operation.

**Strengths:**

- The identification of circuits looks sound and an interesting approach. I enjoyed reading that part of the paper -- thank you.
- The problem being tackled is very interesting, well motivated and some of the experimental results are really insightful


**Weaknesses:**

Main Concerns:
- The task formulation seems severely constrained. For instance, GPT-2 is prompted with the year prefix "XX" already given. Effectively, the approach is evaluating whether GPT-2 can perform 2-digit greater-than operation.
- Even for the restricted case of year-span prediction task, I think it is important to compute correctness of the entire year string, without providing the XX prefix. Would the results hold in that case? I wonder how much probability mass GPT-2 will assign to the correct XX year token. This might significantly alter the "Prob Difference" metric and the results and conclusions.
- The Prob difference metric might also be hiding inaccuracy of GPT-2 in generating the correct year. How often is it the case than GPT-2 assigns high prob to one of the correct answers in its top K tokens?.
- I also think it is important to establish a "prior". What is the default behavior of the model when it is shown even simpler prompts that do not require the generation of a number greater than the given number. i.e. is GPT-2 predisposed to generating monotonically increasing numbers, even when the context doesn't need it to?
- In a similar vein, a control task that prompts the model to generate a smaller number is needed to claim that GPT-2 can indeed perform the great-than operation.
- The other main concern is the fact that each prompt-template yields a somewhat different circuit. The task posed in experiments in Section 5 are the same as the initial setup: predict a 4 digit year given the 2-digit prefix of the correct year. Given this, I do not think the results support the conclusion that gpt-2 has a generalized mechanism to perform greater-than across tasks and contexts.

Minor:
- While "emergent capabilities" have been used as motivation for this work, I am not sure if GPT-2 small's abilities can be referred to as "emergent". From what I understand, they are capabilities of much larger models. Moreover, next token prediction (how the main task in this paper is formulated) is well within the kinds of capabilities GPT-2 small was trained on.

Overall, IMO, this is a very interesting line of work which can be made much stronger and conclusive with some additional experiments and analysis.

**Questions:**

Minor:
- In Fig 5, Did you intend to flip the instance coming from the actual vs 01-dataset?
- Fig 6 is not legible. Please increase the font size
- I don't think I understood what "input residual stream" meant. Please clarify.

**Limitations:**

-

---

> ### Author Rebuttal · Authors · 2023-08-08
>
> Thank you for your thorough review and thoughtful questions!
> > The task formulation is constrained.
>
> True—for why, see Problem Scope in the general response.
> > Compute the correctness of the entire year string, w/o the XX prefix: how much probability mass does GPT-2 assign to the correct XX year token?
>
> When predicting XX, GPT-2 almost always predicts a valid continuation (numeric token >= XX). On average, considering the top-100 tokens, which comprise 0.95 of GPT-2’s output probability mass, valid continuations receive 0.89 (94%) of the probability mass. The year XX is almost always the top token (mean rank 1.125, mean probability: 0.41); most top tokens are other centuries > XX. So GPT-2 generally wants to generate XX or some other valid continuation.
> > Prob diff might hide inaccuracy of GPT-2 in generating the correct year. How often does GPT-2 assign high prob to a correct answer in its top-k tokens?
>
> We find this unlikely, given our high baseline probability difference of 0.81. This indicates that at least 0.81 of all of GPT-2’s probability mass—not just 81% of all probability assigned to numeric tokens—is assigned to a correct token.
>
> However, we also tested this and found that 100% of top-1 tokens, and 98.6% of top-5 tokens are valid (>YY). We thus feel confident that GPT-2 predicts a correct answer with high probability.
> > Is GPT-2 predisposed to generating monotonically increasing numbers, even when it doesn't need to?
>
> Yes. To test this, we gave GPT-2 sequences of 4-digit numbers like “XXY1, XXY2,..., XXYN, XX”. We chose XX, Y1,…,YN randomly for each sequence, respecting tokenization. Sequences were generally not monotone, so any 2-digit continuation YY could have been valid. We found that GPT-2’s behavior depends on the sequence length N. At low N, GPT-2 produces mostly numbers > YN; the proportion of continuations < YN increases smoothly until ~50% by N=20. So even in the context of random sequences, GPT-2 often generates increasing numbers. We found that our greater-than circuit also underlies this behavior.
>
> What does this mean for our study? Greater-than behavior emerges even when no greater-than is required; however, we already knew this, having observed it in the less-than case.
>
> In our study, we acknowledge that GPT-2 performs greater-than in incorrect scenarios, and do not claim to find a mechanism unique to greater-than scenarios. Rather, we want to connect GPT-2’s greater-than behavior to its underlying mechanisms, be they unique or not to the correct contexts. Our claim is that we have found a circuit that computes greater-than regardless of whether GPT-2 ought to perform the greater-than operation. This claim stands in the face of this new evidence.
> > A control task that prompts GPT-2 to generate a smaller number
>
> We test less-than in Sec. 5 of our paper. Given prompts like “17YY is less than 17”, GPT-2 predicts YY. Given “The war ended in 17YY and started in 17”, GPT-2 uses the greater-than circuit to predict a year >YY! So, GPT-2 can perform greater-than but not less-than.
> This suggests that GPT-2’s greater-than ability is not supported by general mathematical understanding; it can’t e.g. use the greater-than circuit to produce less-than by upweighting the opposite tokens. GPT-2’s greater-than circuit is thus in an interesting position: it’s not fully general, but can still be applied in some new contexts.
> > Given that each prompt-template yields a different circuit, and the generalization tasks are the same as the initial setup—predict a 4 digit year given the 2-digit prefix of the correct year—the results don’t support the conclusion that GPT-2 has a generalized mechanism to perform greater-than across tasks and contexts.
>
> We disagree with some aspects of this critique. Not all tasks are about years: we shift the context from years to prices, and monotonically increasing numbers. The circuits are mostly identical except in the last case, where additional components seem to be relevant. However, the number of nodes/edges that differ between the two is small; even our original circuit isn’t a bad fit. We could have quantified this circuit overlap better, and will do so in our revision.
>
> However, we agree that we suggested generalization a bit too strongly. The evidence points more towards a mechanism that can perform greater-than (narrowly defined) in different contexts, but is not necessarily integrated into a broader suite of math capabilities. GPT-2’s greater-than mechanism thus lies between memorization and generalization.
> > I am not sure if GPT-2 small's abilities can be referred to as "emergent". From what I understand, they are capabilities of much larger models.
>
> We share this hesitation. Though Wei et al. list math as an emergent ability in LLMs, GPT-2 is smaller than most LMs claimed to have emergent abilities. We’re happy to change that phrasing—we believe that math abilities in LMs are an interesting phenomenon, emergent or not.
> > In Fig 5, did you intend to flip the instance coming from the actual vs 01-dataset?
>
> Yes! To find the circuit, we corrupt parts of the computational graph using the 01-dataset; those edges that hurt performance when corrupted are part of our circuit. However, during evaluation (Fig. 5), we instead give normal input to the circuit nodes, and 01-input to the other nodes.
>
> The idea is that if the circuit (Fig. 5 center, blue / purple) gets normal input, and the rest of the model (Fig. 5 center, red / purple) gets 01-input, the model should perform the task correctly, as if it were receiving only normal input. This is because the circuit controls model behavior on the task. When we give the circuit 01-input, and the rest of the model normal input, model performance is correspondingly very poor.
> > Meaning of "input residual stream"
>
> The input residual stream to e.g. MLP 10 is the 768-dim input vector that serves as input to MLP 10; this is opposed to the token-level input.
>
> Feel free to send follow-up questions / suggestions!

---

> > ### Comment · Reviewer_Nf2K · 2023-08-18
> > **Thank you for the response**
> >
> > Thanks for providing more details and clarifications. I appreciate this work and studying mathematical abilities of LMs is interesting.
> >
> > My comment re. the scope is not to say that "only greater-than is studied", but that "greater-than has been studied using a handful of prompts". This is why I think the results about generalization of the "greater-than" circuit are not completely supported.
> >
> > That said, I found the rest of the response helpful. I am happy to bump up my score a bit.

---

### Author Rebuttal · Authors · 2023-08-08

We thank the reviewers for their insightful reviews. We're glad you found our problem and approach interesting (Nf2K,GMC8,BwzU,WeSY), our experiments extensive and scientifically sound (BwzU,WeSY), and our paper clear and well-written (eiM2,WeSY). Still, we want to address some key concerns: our choice of task, our contributions, and how our methods and results scale.

## Problem Scope (Nf2K, GMC8, WeSY)
Many reviews noted that we studied a small task. We agree that this greater-than is simple; however, it lies in the middle of a major gap in the interpretability literature: how do LMs implement math abilities?

Our simple task allowed us to dig deep into the model, developing a detailed explanation. While our case study cannot explain math in all LMs, it provides intuition and hypotheses useful across tasks and models. For example, we find that GPT-2 performs greater-than using a mechanism that lies between memorization and generalization; this could explain LMs’ inconsistent math performance more broadly. As circuits techniques mature, these insights could help us tackle larger problems in interpretability.

## Contributions (eiM2,GMC8, BwzU)
Many reviewers were unsure of how our contributions differed from those of other work—we will revise our paper to clarify this. Here, we outline our scientific findings and the methodological contributions that enabled them.

### Scientific Findings
- We find a circuit for greater-than in GPT-2. At submission time, no other causal interpretability work had been done on math in pre-trained LMs. Moreover, there is only one prior work published on circuits in pre-trained LMs [1]; it focuses on a circuit that copies one token from the input. In contrast, our task has a wider output space and richer structure: our circuit must interpret tokens as numerical quantities, and upweight a specific set of tokens not present in the input.

Our circuit is a highly detailed account of an LM’s algorithm; prior work does not make such fine-grained causally motivated claims. This is exemplified by our finding that attention heads pass YY information to MLPs, which then compute greater-than. Although these concrete findings are specific to our circuit, the broad motifs—attention heads moving information into MLPs, which act both directly and indirectly—may generalize. By understanding the mechanisms via which LMs implement specific capabilities, we hope to better understand the overarching mechanisms by which LMs work.

- We show that GPT-2’s implementation of greater-than lies between memorization and generalization. We claim this because in our generalization experiments, the circuit does activate in new greater-than scenarios; however, it cannot support related computations like less-than or equal-to, and doesn’t clearly involve general math representations. It thus reflects neither full math competence nor simple memorization. Through this work, we hope to add nuance to the memorization-generalization dichotomy, and take the first step towards a rich characterization of the states in between them.

We thank reviewers Nf2K and eiM2 for their questions about our circuit’s generalization. We agree that our original framing suggested generalization too strongly, and that this framing better represents both our evidence and how LMs work. We will update the paper.

### Methodological Contributions
Our methods contributions over previous circuits work [1] are as follows:
- Instead of patching individual edges in the model, we patch full subgraphs of our model, allowing us to find a complex circuit with many interconnecting components. This is enabled by our use of rust-circuit [2], a framework that allows for direct manipulation of models’ computational graphs. We did not create rust-circuit, but are the first to use it in this fashion.
- We used separate datasets to conduct our circuit-finding study and to assess the generalization of the hypothesis design on the first dataset, similar to the train/test split used in machine learning. In contrast, preceding LM-circuits work found their circuit and tested it on the same set of prompts.
Finally, some reviewers asked: aren’t many of our methods standard, established methods? Not so: as circuits research is very young, there is not yet a standardized methodology. In writing this paper, we hoped to establish such a toolkit for future circuits researchers by bringing together diverse techniques like the logit lens, PCA on representations, and neuron-level interventions.

## Scaling (GMC8, BwZU, WeSY)
Some reviewers were concerned that our methods or findings would not scale. Regarding circuits methods, automatic circuit-finding [2] is already in development (contemporaneously with our work); moreover, in the time since submission, circuits methods have already been applied to 70B-parameter LMs [3]. Our complex subgraph ablations scale less well; we foresee them being used to zoom in on particular components, rather than to study entire large models.

Our findings, too, have potential to generalize at scale. While our circuit is specific to GPT-2 small, other findings need not be so restricted. Larger models too may coordinate their attention heads/MLPs as we observed, or rely on circuits that only partially generalize. These are only hypotheses; however, via these hypotheses, we lay a foundation for future work that explores these phenomena at scale.

We hope that this rebuttal has addressed your concerns! Please reach out with any other questions—we’d be happy to chat more.

### References
1: Wang et al. 2023. Interpretability in the Wild: a Circuit for Indirect Object Identification in GPT-2 Small. ICLR

2: Goldowsky-Dill et al. 2023. Localizing Model Behavior with Path Patching. ArXiV

3: Conmy et al. 2023. Towards Automated Circuit Discovery for Mechanistic Interpretability. ArXiV

4: Lieberum et al. 2023. Does Circuit Analysis Interpretability Scale? Evidence from Multiple Choice Capabilities in Chinchilla. ArXiV

---

### Decision · Program_Chairs · 2023-09-21

**Decision:**

Accept (poster)

**Comment:**

This paper aims to explain how “greater than” operation is implemented in GPT2-small. They use “path patching”, a causal approach to intervene on activations to discover the circuit in charge of conducting this operation. They use different template variants to verify that the identified circuit is indeed the mechanism used in other contexts that need a “greater than” operation.

All reviewers found the problem interesting, and most had consensus about the scientific validity and the depth of analysis in tackling this problem. The key concern was the generalizability of the findings beyond the GPT2-small model and the “greater than” operation. The authors have tempered their generalization claims, clarified the scope and contributions of this work, and added some empirical evidence to the related questions raised by the reviewers (which I would strongly recommend to be added to the appendix). Most reviewers found the author's response adequately addressing their concerns, and adjusted their rating accordingly after a fruitful constructive discussion.

Despite limitations with respect to generalizability of the findings, this work could be a good addition to the recent literature on mechanistic interpretability work that so far has mostly focused on toy tasks and models (like modular addition and group operations) or other setups like factual associations. It is particularly challenging to find reliable mechanisms across tasks and models in our current LMs, and we need to evaluate work in this space with more calibrated expectations. Therefore, I believe a more thorough investigation in a narrower scope could still be valuable.

In any case, I strongly recommend the authors to take the excellent comments by reviewers into account for future revisions of this work or its extensions.